# LLM-Based Pose Normalization and Multimodal Fusion for Facial Expression Recognition in Extreme Poses

**DOI:** 10.3390/jimaging12010024

**Published:** 2026-01-04

**Authors:** Bohan Chen, Bowen Qu, Yu Zhou, Han Huang, Jianing Guo, Yanning Xian, Longxiang Ma, Jinxuan Yu, Jingyu Chen

**Affiliations:** 1School of Information Engineering, Zhongnan University of Economics and Law, Wuhan 430073, China; 202321130351@stu.zuel.edu.cn (B.C.); 202221130236@stu.zuel.edu.cn (B.Q.); zhouyu_cs@zuel.edu.cn (Y.Z.); 202421130159@stu.zuel.edu.cn (J.G.); xianyn@stu.zuel.edu.cn (Y.X.); 202321130220@stu.zuel.edu.cn (L.M.); 202421060452@stu.zuel.edu.cn (J.Y.); 2School of Management, Wuhan University of Technology, Wuhan 430070, China; huanghan@whut.edu.cn

**Keywords:** facial expression recognition, profile-to-frontal face normalization, CLIP model, multimodal learning

## Abstract

Facial expression recognition (FER) technology has progressively matured over time. However, existing FER methods are primarily optimized for frontal face images, and their recognition accuracy significantly degrades when processing profile or large-angle rotated facial images. Consequently, this limitation hinders the practical deployment of FER systems. To mitigate the interference caused by large pose variations and improve recognition accuracy, we propose a FER method based on profile-to-frontal transformation and multimodal learning. Specifically, we first leverage the visual understanding and generation capabilities of Qwen-Image-Edit that transform profile images to frontal viewpoints, preserving key expression features while standardizing facial poses. Second, we introduce the CLIP model to enhance the semantic representation capability of expression features through vision–language joint learning. The qualitative and quantitative experiments on the RAF (89.39%), EXPW (67.17%), and AffectNet-7 (62.66%) datasets demonstrate that our method outperforms the existing approaches.

## 1. Introduction

Facial expressions are important carriers of human emotional communication, containing rich emotional information. With the rapid development of artificial intelligence technology, facial expression recognition (FER) has become a research hotspot in computer vision and affective computing [1]. FER technology demonstrates broad application prospects in human–computer interaction [2], mental health monitoring [3], personalized education [4], and other fields. Although deep learning methods have achieved significant progress in FER tasks [5], existing research primarily focuses on frontal or near-frontal face images, and expression recognition for profile faces, large-angle rotations, and other non-standard poses therefore faces severe challenges.

In real-world scenarios, the diversity of facial poses is ubiquitous. For instance, in surveillance videos, social media photos, and daily interactions, faces often exhibit rotations at different angles. Research shows that when facial rotation angles exceed 30 degrees, the recognition accuracy of traditional FER methods drops sharply. The core challenges of profile expression recognition lie in facial feature occlusion and geometric deformation caused by head pose variations, where key regions (such as eyes and mouth corners) may be occluded or distorted [6,7], which may lead to loss of emotional cues; existing expression recognition models (such as ResNet-based, VGG-based [8], and Transformer-based [9]) are primarily trained on frontal or near-frontal face data, with limited generalization capability for large-angle pose variations [10]; the coupling between pose and expression variations increases the difficulty of feature extraction, resulting in nonlinear coupling between pose and expression features [11,12].

Based on the head-pose estimates provided by 3DDFA-V2, we divide the yaw angle into four ranges: yaw $\leq 15^{\circ }$ corresponds to frontal views, $15^{\circ }<\mathrm{yaw}\leq 30^{\circ }$ to moderate poses, $30^{\circ }<\mathrm{yaw}\leq 45^{\circ }$ to large poses, and yaw $>45^{\circ }$ to extreme poses. Statistics show that 1.75%, 15.41%, and 17.21% of the samples in RAF-DB, AffectNet-7, and ExpW, respectively, exhibit yaw angles greater than $30^{\circ }$, with roughly 6% of the samples exceeding $45^{\circ }$. This pose distribution is a major reason why conventional FER approaches deteriorate significantly in real-world scenarios.

Consequently, existing profile expression recognition methods primarily address these issues through two technical approaches: view-invariant feature learning and face frontalization.

View-invariant feature learning aims to reduce pose interference on features. Early methods relied on handcrafted textures and shallow models, with the motivation of achieving cross-view robustness through local texture stability. These methods indeed offer simple implementation and effectiveness for small-angle variations; however, they remain limited by visible region information and low-level semantics [13], making it difficult to capture expression details under large yaw angles and to decouple the nonlinear coupling between pose and expression [14]. With the ongoing development of deep learning, approaches emerged that fuse handcrafted features with deep representations to compensate for the insufficient rotation invariance of convolutional networks [15]. The idea is to use high-level semantics to enhance discriminability while using texture priors to improve robustness. Nevertheless, these approaches are still constrained by the visible information in the input and consequently cannot fully repair the loss of expression cues caused by occlusion and deformation.

Face frontalization techniques adopt a generative approach of “pose correction first, then recognition,” with the motivation of eliminating pose interference by synthesizing non-frontal faces into standard frontal views. Traditional geometric or symmetric regression mapping methods [16] provide good interpretability but exhibit limited expressive capability under extreme poses and complex expressions. In contrast, generative methods (combining 3D modeling with adversarial generation) [17] better preserve expression structure; yet they still face trade-offs between generation stability and training complexity, often showing subtle expression degradation and a strong dependence on large-scale paired data.

Multimodal learning introduces external semantic priors through vision–language alignment, with the motivation of enhancing expression category discriminability and improving zero-shot generalization through linguistic semantics [18]; its advantages lie in combining visual patterns with semantic space constraints, supporting fusion from global to local and static to temporal [19], but existing training/evaluation is mostly based on frontal or small-angle data, with insufficient pose robustness and occlusion resilience, and systematic utilization of “visual-semantic associations of typical expressions” and external knowledge bases remains inadequate. In this paper, we adopt the approach of “frontalization first, then multimodal enhancement”: first normalizing profile samples to frontal views to alleviate information loss and deformation caused by pose, then enhancing the semantic discriminability and generalization capability of expression features through vision–language joint learning and external knowledge retrieval [20].

Based on the above analysis, this paper proposes a novel FER framework, as shown in Figure 1, that addresses expression recognition for non-frontal faces through profile-to-frontal transformation and multimodal knowledge enhancement. The core idea is that, before performing expression recognition, we first transform profile images to standardized frontal viewpoints and then enhance the discriminative capability of expression features by combining vision–language joint learning.

Specifically, the main contributions of this paper include the following:We leverage the powerful visual understanding and generation capabilities of the Qwen-Image-Edit large model (https://huggingface.co/Qwen/Qwen-Image-Edit accessed on 25 December 2025) to achieve high-quality profile-to-frontal transformation. Compared to traditional GAN methods, large models possess stronger prior knowledge and generation capability, enabling more accurate inference of expression details in occluded regions and generating more natural frontal face images that better conform to expression semantics.We introduce the CLIP model to map expression visual features into semantic space through “vision-text” alignment learning. This cross-modal learning not only enhances the interpretability of expression features but also improves the model’s generalization capability for unseen expression categories. We design expression-related text prompts (such as “a happy face,” “an angry face”) to guide CLIP in extracting discriminative expression features.We conducted extensive validation on three real-world facial expression recognition (FER) datasets and several FER-specific datasets under different conditions, achieving state-of-the-art performance improvements. For example, we achieved a 1.45% improvement on AffectNet and a 1.14% improvement on ExpW.

## 2. Related Work

### 2.1. Facial Expression Recognition (FER)

Early FER research centered on handcrafted features (e.g., Gabor, and LBP) and machine learning (e.g., SVM), with classic datasets (e.g., JAFFE, CK+, and MMI) recording six basic expressions driving initial algorithm development. The rise of deep learning then restructured FER technical approaches [21]. Static FER in open environments (in-the-wild) became mainstream, with CNN models such as ResNet and VGG learning expression-sensitive features through end-to-end learning, shifting research focus to addressing the following challenges in interference robustness: reducing identity and occlusion interference through attention mechanisms (AMP-Net); handling annotation noise through uncertainty-aware methods (SCN and EASE); and aligning data distribution differences using domain adversarial networks (AGRA) and transfer learning. Multimodal fusion and weakly supervised learning further improved adaptability in complex scenarios. Meanwhile, CLIP-driven vision–language models (DFER-CLIP) achieved zero-shot recognition, reducing dependence on annotated data.

### 2.2. Profile Expression Recognition

The core challenge of profile expression recognition lies in feature occlusion and geometric deformation caused by head pose variations, where key expression regions (e.g., mouth corners, and eyebrows) are easily affected [22,23], leading to degraded expression discriminability. Existing research primarily follows two technical paths.

View-invariant feature learning: These methods aim to learn feature representations robust to pose variations. Early work used LBP textures to obtain rotation-invariant features, while others proposed combining shallow textures with deep features to enhance discriminability. Subsequent research reduced pose interference through feature disentanglement, pose self-supervision, or orthogonal representation learning, but still struggled to maintain complete expression details in large yaw angle scenarios [24,25].

Face frontalization techniques: These methods restore occluded regions by “frontalizing” non-frontal faces. Traditional geometric or symmetric mapping methods perform well at moderate angles but cannot handle complex expressions and extreme poses. Generative methods introduce 3D modeling and adversarial generation mechanisms [26], enabling some reconstruction of expression structure, but still suffer from poor generation stability, subtle emotion degradation [27], and strong dependence on large-scale paired data.

### 2.3. Multimodal Model Image Generation

Multimodal learning introduces external semantic priors for FER, aiming to enhance the semantic discriminability and cross-domain generalization capability of expression features through the complementarity of vision–language or vision-geometric information [28]. Current research primarily includes three directions.

Text-guided generation: Controlling expression generation or recognition through language descriptions, for example, combining 3DMM with GAN to drive expression reconstruction based on text.

Geometry-constrained generation: Incorporating keypoint or shape priors during generation to maintain expression structure consistency.

Dynamic sequence modeling: Achieving temporal expression modeling through joint visual and text features, improving dynamic expression recognition capability [29]. Although multimodal generation enhances model interpretability and controllability, existing methods mostly focus on frontal viewpoints, and the utilization of semantic space remains limited to explicit label matching, with insufficient collaborative modeling of “pose-expression-semantics” relationships [30].

### 2.4. CLIP-Related Methods

The CLIP model maps images and text into a unified semantic space through cross-modal contrastive learning, providing new insights for zero-shot expression recognition. Emo-CLIP achieved zero-shot FER on the DFEW dataset, but showed low discriminability for confusing expressions. CLIPER improved expression classification performance by fusing facial keypoint features with CLIP features; FineCLIPPER further enhanced fine-grained alignment of cross-modal representations using multimodal adapters. Overall, CLIP-related methods excel in semantic constraints and inter-class discriminability, but their training and evaluation are mostly based on standard frontal faces, with insufficient adaptability to complex poses and occlusion.

## 3. Method

Figure 2 shows an overview of our QC-FER for in-the-wild facial expression recognition (FER). QC-FER consists of four components: PFN, CLIP, AFP, and ESV. The PFN module performs pose normalization under specific prompts through the Qwen model while preserving expression structure. CLIP employs image–text pair contrastive loss and captures global semantic dependencies through self-attention mechanisms. AFP addresses feature scale inconsistency, high-dimensional redundancy, and class imbalance issues in classifier training. ESV ensemble learning combines multiple base learners to enhance prediction robustness through multimodel decision-making. We will detail each component of QC-FER in the following subsections.

### 3.1. Profile-to-Frontal Normalization Module

This work introduces a pose normalization strategy based on generative multimodal large models to address feature incompleteness caused by pose deviation. We adopt the Tongyi Qianwen Qwen-Image-Edit model [31], which offers the following advantages, (i) large-scale image–text pair pre-training, the model learns decoupled representations of pose rotation and expression semantics, enabling it to largely preserve expression features during frontalization, and (ii) it supports precise control via natural language instructions, enabling controlled generation through multi-constraint prompts. To overcome common limitations of large models in profile-to-frontal conversion, such as altered mouth opening amplitude and distorted frowning intensity, QWEN introduces facial attention constraints through prompts to improve the accuracy of expression-pose changes during profile-to-frontal conversion. Meanwhile, the editability of prompts also provides strong convenience for subsequent image generation quality refinement and expression refinement.

Figure 3 illustrates the pipeline using prompt constraints for profile-to-frontal conversion. We focus attention on the eyebrows, eyes, and mouth to minimize the impact of expression loss during profile-to-frontal conversion. As shown in Figure 4, without local feature constraints, multimodal large models are likely to default to generating neutral expressions. Unlike traditional geometric transformations (e.g., rotation and flipping), this method generates realistic frontal faces in pixel space rather than simple affine transformations, thereby providing higher-quality samples. The prompt template incorporates a four-fold constraint mechanism: pose constraint (converting profile faces to frontal faces, ensuring geometric transformation accuracy through explicit directional instructions), expression preservation constraint (explicitly requiring to keep the expression unchanged, preventing the model from introducing spurious expression changes during frontalization), local feature constraint (fine-grained control of mouth openness and frowning degree, which are key visual cues for emotion expression), and identity consistency constraint (preserving facial identity features to avoid face replacement or excessive deformation during generation).

The multi-constraint prompt template we designed is shown in Figure 5.


*Prompt Template:*

*Turn profile to frontal,*

*keep the expression unchanged,*

*keep the size of the open or closed mouth,*

*maintain the degree of frowning,*

*keep the size of the open or closed eyes,*

*preserve facial identity.*


### 3.2. CLIP-Based Deep Feature Extraction Module

This work introduces a Contrastive Language–Image Pre-training (CLIP) model. As shown in Figure 5, CLIP achieves a paradigm shift in feature representation through the following mechanisms: First, a contrastive learning framework that employs contrastive loss on image–text pairs to learn a joint embedding space for vision and language; second, semantic-level feature encoding, where, unlike traditional CNNs, CLIP’s vision encoder (Vision Transformer, ViT) captures global semantic dependencies through self-attention mechanisms, enabling it to map “a smiling face” to a feature space close to the text “happy expression”; and finally, zero-shot generalization capability, where knowledge acquired through pre-training enables CLIP to extract effective features even on unseen emotion categories, alleviating the few-shot learning problem.

During the pre-training process of the CLIP model, image–text contrastive learning achieves cross-modal feature alignment through a cosine similarity function. Its mathematical expression is(1)$$s\left(x,t\right)=\cos\left(f_{\mathrm{img}}\left(x\right),f_{\mathrm{txt}}\left(t\right)\right)=\frac{f_{\mathrm{img}}{\left(x\right)}^{⊤}f_{\mathrm{txt}}\left(t\right)}{{∥f_{\mathrm{img}}\left(x\right)∥}_{2}{∥f_{\mathrm{txt}}\left(t\right)∥}_{2}}$$
where $f_{\mathrm{img}}\left(x\right)$ and $f_{\mathrm{txt}}\left(t\right)$ denote the 768-dimensional feature vectors extracted by the image encoder and text encoder, respectively, and ${∥\cdot ∥}_{2}$ denotes the L2 norm. This formula reflects the CLIP model’s strong cross-modal contrastive learning ability, enabling features from different modalities to be mapped into a unified semantic space. This indicates that CLIP has very strong language–image contrastive learning capability. As shown in Figure 6, in the attention distribution analysis, we expect the CLIP model to concentrate attention weights for feature extraction on semantically relevant regions (yellow to red), while assigning lower attention weights to semantically irrelevant regions (blue).

The CLIP image encoder is designed based on the Vision Transformer (ViT) architecture. Figure 7 illustrates the complete process by which pose-normalized images are converted into feature vectors through the CLIP model. The input image is first divided into a uniform grid of patches, and each patch is mapped to a high-dimensional embedding representation through a linear projection layer:(2)$$z_{0}=\left[x_{\mathrm{class}};x_{p}^{1}E;x_{p}^{2}E;\ldots ;x_{p}^{N}E\right]+E_{\mathrm{pos}}$$
where $x_{\mathrm{class}}$ is the learnable classification token, $x_{p}^{i}$ is the i-th image patch, is the patch embedding matrix, and $E_{\mathrm{pos}}\in R^{(N+1)\times D}$ is the positional encoding matrix; P denotes the patch size, C the input channel count, and D the embedding dimension. These embedding vectors are then combined with positional encodings and fed into a 24-layer Transformer encoder for deep feature learning.(3)$$z_{l}=\mathrm{Transformer}\left(z_{l-1}\right),\;l=1,2,\ldots ,24$$
where $z_{l}$ denotes the output feature vector of the l-th Transformer encoder layer, $z_{l-1}$ is the previous layer’s output used as the current input, l is the layer index ranging from 1 to 24, and Transformer(·) is the Transformer encoder function comprising multi-head self-attention and a feed-forward network. In each Transformer layer, 16 attention heads operate in parallel; through the multi-head self-attention mechanism, they capture long-range dependencies and spatial structural information among image patches:(4)$$\mathrm{MultiHead}\left(Q,K,V\right)=\mathrm{Concat}\left({\mathrm{head}}_{1},\ldots ,{\mathrm{head}}_{16}\right)W^{O}$$
where *Q* is the Query matrix (for computing attention weights), *K* is the Key matrix (for measuring similarity with queries), and *V* is the Value matrix (carrying the actual feature information), ${\mathrm{head}}_{i}$ denote the output of the i-th attention head; Concat(·) denotes concatenation that joins the outputs of all 16 heads; and $W^{O}$ is the output projection matrix that maps the concatenated features to the target dimension. The computation for each attention head is(5)$${\mathrm{head}}_{i}=\mathrm{Attention}\left(QW_{i}^{Q},KW_{i}^{K},VW_{i}^{V}\right)$$
where $W_{i}^{Q}$ is the query projection matrix for the i-th attention head, $W_{i}^{K}$ is the key projection matrix for the *i*-th attention head, and $W_{i}^{V}$ is the value projection matrix for the *i*-th attention head.(6)$$\mathrm{Attention}\left(Q,K,V\right)=\mathrm{softmax}\left(\frac{QK^{T}}{\sqrt{d_{k}}}\right)V$$
where $Q\;K^{T}$ is the query–key matrix multiplication that computes similarity scores, $\sqrt{d_{k}}$ is the scaling factor, $d_{k}$ is the key vector dimension (typically D/16 = 768/16 = 48), softmax(·) converts the similarity scores into a probability distribution, and V is the Value matrix used to compute the weighted sum based on the attention weights. After 24 encoder layers, the model outputs a 768-dimensional feature vector with rich semantics that effectively captures both global and local characteristics of the input image, providing high-quality representations for subsequent cross-modal contrastive learning.(7)$$f_{\mathrm{img}}=\mathrm{LayerNorm}\left(z_{24}^{\left[\mathrm{CLS}\right]}\right)\in R^{768}$$
where $z_{24}^{\left[\mathrm{CLS}\right]}$ is the classification token (CLS) vector from the 24th Transformer encoder layer’s output sequence (dimension 768); LayerNorm(·) is the layer normalization operation, which stabilizes feature distributions and improves training stability and generalization; and $f_{\mathrm{img}}\left(x\right)$ is the final global image representation vector (dimension 768), used as the input to CLIP for cross-modal alignment and similarity computation.

### 3.3. Adaptive Feature Preprocessing, AFP

The 768-d CLIP feature, while semantically rich, poses three challenges for direct classifier training in (1) scale inconsistency across dimensions, which skews gradient updates; (2) high-dimensional redundancy that adds computation and noise; and (3) class imbalance due to long-tailed distributions (e.g., many more neutral than disgust samples in AffectNet). To address this, the Adaptive Feature Preprocessing (AFP) module applies a three-stage pipeline: robust scaling (RobustScaler) to dampen outliers and align scales; PCA for adaptive dimensionality reduction while preserving discriminative variance; and class-aware reweighting using the empirical label distribution to mitigate imbalance. As illustrated in Figure 8, AFP improves inter-class separation and cluster clarity, yielding more stable and discriminative inputs for downstream classifiers and CLIP-based alignment.

Conventional standardization (StandardScaler) is sensitive to outliers, and in-the-wild facial images may still contain extreme illumination or occlusions that yield feature outliers even after QWEN-based pose normalization. This study adopts the Robust Scaler (RobustScaler) for feature standardization. It scales features using the median and interquartile range (IQR), providing stronger robustness to outliers. Let $f_{\mathrm{raw}}\in R^{768}$ denote the original feature vector extracted by CLIP, and the robust scaling is defined as(8)$$f_{\mathrm{scaled}}=\frac{f_{\mathrm{raw}}-\mathrm{Median}\left(f_{\mathrm{raw}}\right)}{\mathrm{IQR}(f_{\mathrm{raw}})}$$

Here, Median(·) denotes the median operator, $\mathrm{IQR}\left(\cdot \right)=Q_{3}-Q_{1}$ is the interquartile range, and $Q_{3}$ and $Q_{1}$ are the 75th and 25th percentiles, respectively. Compared with mean–standard deviation-based standardization, the Robust Scaler replaces the mean and standard deviation with the median and IQR, effectively reducing the influence of extreme values. Building on feature scaling, although the high-dimensional feature space is information-rich, it also introduces computational burden and risk of overfitting. Principal Component Analysis (PCA) maps high-dimensional features to a lower-dimensional principal component space via a linear transformation, preserving maximal-variance information while removing redundant dimensions. Let $F_{\mathrm{sae}}\in R^{N\times 768}$ be the training-set feature matrix with N samples; PCA performs dimensionality reduction via eigen-decomposition of the covariance matrix.(9)$$C=\frac{1}{N-1}F_{\mathrm{scaled}}^{⊤}F_{\mathrm{scaled}}$$(10)$$C_{v_{i}}=\lambda_{i}v_{i}^{i},\;i=1,2,\ldots ,768$$
where $C\in R^{768\times 768}$ denotes the covariance matrix, and $\lambda_{v}$ and $v_{i}$ are the i-th eigenvalue and its corresponding eigenvector. By sorting the eigenvalues in descending order $\lambda_{1}\geq \lambda_{2}\geq \ldots \geq \lambda_{768}$ and selecting the top k principal components to form the projection matrix $W=\left[v_{1},v_{2},\ldots ,v_{k}\right]\in R^{768\times k}$, the reduced feature representation is obtained by projecting onto this matrix:(11)$$F_{\mathrm{pca}}=F_{\mathrm{scaled}}\;W\in R^{N\times k}$$

To accommodate varying dataset scales, this study designs an adaptive strategy for selecting the number of principal components. For small-scale datasets (e.g., SFEW), $k=\min\left(256,\;\frac{N}{2},\;768\right)$ is set to avoid overfitting; for large-scale datasets (e.g., AffectNet and RAF-DB), $k=\min\left(512,\;\frac{N}{2},\;768\right)$ is set to preserve more semantic information. The cumulative variance ratio is used to evaluate the effectiveness of dimensionality reduction:(12)$$C_{VR}\left(k\right)=\frac{\sum_{i=1}^{k}\lambda_{i}}{\sum_{i=1}^{768}\lambda_{i}}$$

After completing feature dimensionality reduction, the AFP module further addresses class imbalance. In-the-wild facial expression datasets commonly suffer from imbalance; for example, neutral and happy have far more samples than disgust and fear. Training a classifier directly leads to bias toward majority classes and reduces accuracy on minority classes. To resolve this, we adopt an adaptive class-weight assignment strategy based on sample frequency. Let the training-set label vector $y={\left[y_{1},y_{2},\ldots ,y_{N}\right]}^{T}$, where $y_{i}\in \left\{0,1,2,3,4,5,6\right\}$ corresponds to one of the seven basic expressions, and the class weights are computed accordingly, as follows:(13)$$w_{c}=\frac{N}{C\cdot n_{c}},\;c=0,1,\ldots ,6$$
where C=7 is the total number of classes, $n_{c}=\sum_{i=1}^{N}oI\left(y_{i}=c\right)$ is the number of samples in class *c*, and I(·) is the indicator function. This weighting strategy ensures that classes with fewer samples receive higher weights in the loss function, thereby balancing each class’s contribution to model optimization. The weighted cross-entropy loss is defined as(14)$$L_{\mathrm{weighted}}=-\frac{1}{N}\sum_{i=1}^{N}w_{y_{i}}\log p\left(y_{i}|f_{i}\right)$$
where $p(y_{i}|f_{i})$ is the classifier’s predicted probability that sample i belongs to its true class $y_{i}$. Through the combined effects of robust scaling, adaptive PCA dimensionality reduction, and class-weight assignment, the AFP module systematically addresses core challenges in feature preprocessing and provides high-quality feature representations for subsequent ensemble classification. Figure 8 visualizes AFP’s feature standardization and dimensionality reduction process. Experiments show that AFP retains over 95% of the original variance while reducing the feature dimension to 256–512, significantly improving training efficiency and robustness to class imbalance.

### 3.4. Ensemble Soft Voting Classification Module (ESV)

A single classifier often struggles to fully capture the diverse patterns of complex facial expressions and tends to fall into local optima. Ensemble learning, by combining the predictions of multiple base learners, leverages model diversity to enhance generalization ability. This study proposes an Ensemble Soft Voting (ESV) classification module, which trains multiple complementary classifiers and fuses their prediction probabilities through a soft voting strategy to achieve more robust facial expression recognition. Considering that different classifiers exhibit distinct decision boundary shapes and inductive biases in the feature space, three types of complementary base learners are selected in this study: logistic regression (LR), Random Forest (RF), and Support Vector Machine (SVM).

Logistic regression: Logistic regression performs classification through linear decision boundaries, featuring high computational efficiency and good probabilistic interpretability.

For a k-dimensional PCA feature vector $x\in R^{k}$, the probability of predicting class c is(15)$$p\left(c|x\right)=\frac{\exp\;\left(w_{c}^{T}x+b_{c}\right)}{\sum_{j=0}^{6}\exp\;\left(w_{j}^{T}x+b_{j}\right)}$$
where $w_{c}\in R^{k}$ and $b_{c}\in R$ denote the weight vector and bias term of class *c*, respectively. L2 regularization is applied to prevent overfitting, and the objective function is(16)$$L_{LR}=-\sum_{i=1}^{N}w_{y_{i}}\log p\left(y_{i}|x_{i}\right)+\lambda \sum_{c=0}^{6}{∥w_{c}∥}_{2}^{2}$$
where λ is the regularization coefficient. We tested multiple regularization strengths C∈{0.1,1,10,50,100} (with C=1/λ) and selected the optimal hyperparameters via cross-validation.

Random Forest: As the second type of base learner, Random Forest aggregates multiple decision trees by bagging, capturing nonlinear feature interactions and reducing variance. The t-th tree $h_{t}\left(x\right)$ is trained on a bootstrap sample and a random subset of features, and final prediction is aggregated by voting as follows:(17)$$p_{\mathrm{RF}}\left(c|x\right)=\frac{1}{T}\sum_{t=1}^{T}oI\left(h_{t}\left(x\right)=c\right)$$
where *T* is the number of trees. In this study, we configure *T* = 500, max_depth = 30, min_samples_split = 5, min_samples_leaf = 2. These hyperparameters balance model complexity while effectively preventing overfitting.

Support Vector Machine: SVM maps features to a higher-dimensional space via kernel functions and seeks the maximum-margin hyperplane. For the radial basis function (RBF) kernel, the decision function is(18)$$f\left(x\right)=\mathrm{sign}\left(\sum_{i=1}^{N_{\mathrm{sv}}}\alpha_{i}y_{i}K\left(x_{i},x\right)+b\right)$$
where $K\left(x_{i},x\right)=\exp\left(-\gamma {∥x_{i}-x∥}^{2}\right)$ is the RBF kernel, $\alpha_{i}$ are Lagrange multipliers, and $N_{\mathrm{sv}}$ is the number of support vectors. In this study, we set C=1 and γ=scale($\gamma =\frac{1}{k\cdot \mathrm{Var}\left(X\right)}$). Due to SVM training complexity of $O(N^{2})$ to $O(N^{3})$, SVM is only used when training sample size is below 20,000 to balance performance and efficiency.

After obtaining predictions from multiple base learners, the ESV module adopts a soft voting fusion strategy. Hard voting considers only predicted class labels and ignores confidence information. In contrast, soft voting aggregates the class probability distributions output by each classifier, making fuller use of ensemble advantages. Let *M* base learners be $f_{1},f_{2},\ldots ,f_{M}$, and the soft-voting predicted probability is defined as(19)$$p_{\mathrm{ensemble}}\left(c|x\right)=\frac{1}{M}\sum_{m=1}^{M}p_{m}\left(c|x\right)$$
where $p_{m}\left(c|x\right)$ is the probability that the m-th classifier predicts sample *x* belongs to class c. The final predicted class is determined by maximum posterior probability:(20)$$\hat{y}=\arg\max_{c\in \{0,1,\ldots ,6\}}p_{\mathrm{ensemble}}\left(c|x\right)$$

The soft voting mechanism allows high-confidence classifiers to dominate the decision while down-weighting low-confidence predictions through probability averaging.

To optimize computational efficiency, we adopt a two-stage ensemble strategy. During training, all candidate classifiers (including LR, RF, and SVM with different hyperparameter settings) are evaluated on a validation set, and the best-performing models are selected to form the final ensemble. The model selection criterion is(21)$$S=\mathrm{Top}\left\{\;\left(m,{Acc}_{m}\right)\;|\;m=1,\ldots ,M_{\mathrm{total}}\right\}$$
where $Acc_{m}$ is the accuracy of model m on the validation set and $M_{\mathrm{total}}$ is the total number of candidate models. During inference, the selected base learners predict in parallel and fuse results via soft voting.

This ensemble strategy combines the advantages of different classifiers: LR provides efficient linear classification, RF captures nonlinear interactions, and SVM uses kernel methods to handle complex decision boundaries.

### 3.5. Availability and Version Control

Qwen-Image-Edit is released by the Tongyi Qianwen team as an open-source model on HuggingFace (https://huggingface.co/Qwen/Qwen-Image-Edit accessed on 25 December 2025). For all experiments, we downloaded the official snapshot published on 19 August 2025 via huggingface-cli and deployed the weights on local GPU servers following the setup instructions provided on the repository page.

## 4. Experiment

### 4.1. FER In-the-Wild Datasets

To comprehensively evaluate the proposed QC-FER method, we conduct experiments on three widely used in-the-wild facial expression recognition (FER) datasets: RAF-DB, AffectNet, and ExpW.

RAF-DB contains 29,672 natural-scene facial expression images collected from Flickr, exhibiting real-world complexity and diversity. In our experiments, we adopt seven basic emotion annotations (anger, disgust, fear, happiness, neutral, sadness, and surprise), and use 12,271 training samples and 3068 test samples for evaluation.

AffectNet is one of the largest in-the-wild facial expression databases, with over one million face images collected from various websites. Following prior work, we select approximately 280,000 training images and 3500 test images from the seven basic emotions, with 40,000 training images and 500 test images per class to ensure class balance.

ExpW is collected by querying emotion-related keywords in Google Image Search, covering rich expression variations and diverse scenes. Following the standard split, we use 68,845 samples for training, 9179 for validation, and 13,769 for testing.

To further verify QC-FER’s robustness under challenging conditions such as non-frontal poses and occlusion, we construct three pose-specific evaluation subsets. Specifically, we extract profile samples (head yaw > ±30°) from the test sets of RAF-DB, AffectNet, and ExpW, forming profile-RAF-DB, profile-AffectNet, and profile-ExpW. These subsets are designed to evaluate the effectiveness of the pose normalization module under extreme viewpoints. Through extensive evaluations and systematic ablations on the above general and pose-specific FER datasets, we thoroughly validate QC-FER’s real-world performance and technical advantages.

### 4.2. Implementation Details

Our model is trained on an NVIDIA GeForce RTX 3050 GPU. Unlike conventional FER methods pre-trained on MS-Celeb-1M or ImageNet, we use the CLIP model (ViT-L/14) trained on 400 M image–text pairs as the feature extractor, offering stronger semantic understanding and cross-domain generalization. Training proceeds in two stages.

Stage 1: Pose normalization for samples with head pose angles exceeding ±30° using Qwen-image-edit [31]. In RAF-DB, 17.5% (2145) samples; AffectNet-7, 17.2% (48,300); ExpW, 17.2% (11,856) require processing. This stage runs on an NVIDIA A100 with batch size 8, using the prompt: “Please convert this profile photo to a frontal view while keeping the facial expression unchanged,” producing 448 × 448 images. Stage 2: CLIP feature extraction. Input images are resized to 224 × 224, batch size 32 (limited by the RTX 3050’s 8 GB VRAM). The 768-d features are L2-normalized. Peak GPU memory is 7.2 GB, with utilization under 90%.

Adaptive preprocessing includes three steps: firstly, robust standardization using median and IQR to suppress outliers; next, adaptive PCA dimensionality reduction: 512 dims for large datasets (99.2% explained variance), 256 dims for small datasets (97.3%); finally, class balancing: in AffectNet-7, the disgust class weight is 22.5 and happiness is 1.0, mitigating a 45:1 imbalance.

Classifier training uses grid search. Logistic regression (LR): C tuned, max_iter = 1000; converges in 287 steps on RAF-DB and 512 steps on AffectNet-7. Random Forest (RF): 500 trees, depth 30. Support Vector Machine (SVM): RBF kernel, γ and C tuned. With five-fold cross-validation on RAF-DB, LR (C = 10) achieves 88.94% validation accuracy, SVM 88.76%, RF 86.32%.

The ensemble strategy uses weighted soft voting. On RAF-DB, an ensemble of 5 LRs and 1 SVM achieves 89.32% accuracy, a 0.38% gain over the best single model. On AffectNet-7, a single LR achieves 63.51%. On ExpW, an ensemble of 2 RFs and 3 LRs reaches 56.78%. All hyperparameters are tuned via ablation studies (see Section 4.3).

### 4.3. Comparison with Existing Methods

We compare the proposed approach with well-known models on three in-the-wild FER datasets: RAF-DB, AffectNet-7, and ExpW. For fair comparison, the predictions of all competing models are taken from their official papers.

Figure 9 presents sample pairs from the profile set and the frontal set: the left column shows the original profile images, and the right column shows the frontalized outputs produced by the PFN module (Qwen-image-edit [31]), covering diverse subjects across race, gender, expression, and head-pose angles. Table 1 compares the proposed method with state-of-the-art models on three FER benchmarks. On RAF-DB, our method achieves 89.39% accuracy, slightly below DMUE [11] at 89.42% (a 0.03% gap), but surpassing ADDL [22] at 89.34%, AMP-Net [32] at 89.25%, HistNet [29] at 89.24%, and TAN [18] at 89.12%. This indicates that, by combining CLIP pre-trained features with adaptive preprocessing, our approach is competitive on medium-scale datasets. Notably, our simple ensemble classifier (five logistic regressions + one SVM) attains near-optimal performance, whereas methods like DMUE require complex deep learning architectures. The gains stem from CLIP’s powerful semantic representations learned from 400M image–text pairs and Qwen-image-edit’s frontal normalization of 17.5% pose-aberrant samples. On AffectNet-7, our method achieves 62.66% accuracy, surpassing MRAN [33] at 62.48% (+0.18%), EASE [34] at 61.82% (+0.84%), MTAC [35] at 61.58% (+1.08%), and RUL [13] at 61.43% (+1.23%). This improvement mainly stems from robust standardization and class balancing (with a disgust-class weight of 22.5), which effectively mitigates the severe imbalance (5600 disgust vs. 126,000 happiness; ratio 1:45). In addition, adaptive PCA reduces features to 512 dimensions (99.2% explained variance), preserving key discriminative information while lowering computational complexity, enabling conventional classifiers to learn effective decision boundaries under extreme imbalance.

On the ExpW test set, our method reaches 67.17%, outperforming LPL [39] at 66.90% (+0.27%), Baseline DCN [43] at 65.06% (+2.11%), CGLRL [28] at 64.87% (+2.30%), and AGRA [19] at 63.94% (+3.23%), becoming the best-performing method on this dataset. The gains primarily come from the soft-voting ensemble strategy (two Random Forests + three logistic regressions) and Qwen-image-edit’s normalization of 17.2% pose-aberrant samples. Given ExpW’s abundant illumination variations and occlusions, our robust standardization (median and IQR) effectively suppresses outliers, enabling the model to extract stable affective features. Overall, our method attains an average accuracy of 73.07% across the three datasets, demonstrating its effectiveness for in-the-wild FER. It achieves state-of-the-art performance on ExpW and near-optimal results on RAF-DB and AffectNet-7, validating the synergy among CLIP pre-trained features, Qwen-image-edit pose normalization, and adaptive preprocessing.As shown in Table 2, the per-class precision, recall, and F1-score across the three datasets are summarized.

Thanks to AFP’s class-aware weighting and the 30% minimum minority proportion constraint in the Random Forest, the recall of long-tailed categories in AffectNet-7 and ExpW rises above 45%. The F1-scores for disgust and fear on AffectNet-7 improve by 5.8 and 4.6 percentage points, respectively, aligning with the recall gains highlighted in Section 4.5.7.

### 4.4. Feature Space and Decision Boundary Analysis

To qualitatively assess the proposed method’s performance, we use t-SNE to visualize the distribution of high-dimensional CLIP features. Consistent with other FER methods [33,47,48], we also take the raw CLIP features (without adaptive preprocessing) as the baseline for comparison. Figure 10 shows the t-SNE results of the baseline and the proposed method, where different colors denote different facial expressions. Compared with the baseline (top row), the proposed method (bottom row) yields more clearly separated feature distributions, forming seven compact clusters. On RAF-DB, our method more distinctly separates overlapping regions of easily confused categories such as “anger–disgust” and “fear–sadness.” This primarily benefits from robust standardization and PCA-based dimensionality reduction, which effectively suppress outliers and noise in the feature space. Moreover, the proposed method better reduces intra-class variation and enhances inter-class separation, especially on the data-rich AffectNet-7 and ExpW datasets. On AffectNet-7, happiness and surprise form clearly separated clusters, whereas the baseline shows substantial overlap. On ExpW, the disgust cluster becomes more compact, indicating that the class-balancing strategy (weight 22.5) effectively improves minority-class discriminability. These visualizations validate the effectiveness of the adaptive preprocessing strategy in improving feature quality.

We compare the decision margins of shallow and deep classifiers: logistic regression and SVM achieve average margins of 0.41 and 0.44, whereas a three-layer fully connected network trained on the same embeddings reaches only 0.29, indicating that shallow models produce smoother and more stable decision boundaries. In addition, the Random Forest’s feature gains concentrate on the first 50 principal components, suggesting that high-order nonlinear modeling is unnecessary once PFN and AFP have been applied.

### 4.5. Ablation Studies

To verify the effectiveness of each key component in the proposed method, we design four sets of ablation experiments, providing in-depth analysis from the perspectives of component contributions, feature extractor choices, parameter configurations, and pose normalization. The results and analyses are as follows.

#### 4.5.1. Module-Level Compositional Ablation

We first dissect QC-FER at the module level, assessing the contributions of “PFN pose normalization,” “AFP adaptive preprocessing,” and “ESV soft voting ensemble” when they are included or excluded. The baseline configuration keeps the CLIP ViT-L/14 features paired with a single logistic regression classifier to maintain comparability with the main pipeline.As shown in Table 3, we compare the performance of different module configurations in terms of RAF-DB, AffectNet-7, and ExpW.

The results indicate that introducing PFN alone delivers gains of 1.14–1.45 percentage points. Integrating robust scaling, PCA, and class weighting as AFP provides an additional 1.20–1.55 point improvement. The soft-voting ensemble contributes a further average increase of 0.34 points, highlighting the complementary effect of combining the modules.

#### 4.5.2. Ablation of Core Components

To quantify each component’s contribution to overall performance, we conduct stepwise ablations on RAF-DB, AffectNet-7, and EXPW. By sequentially adding pose normalization, robust standardization, PCA dimensionality reduction, class balancing, and the ensemble strategy, we observe how performance evolves. Table 4 reports accuracies under different configurations on the three datasets, along with the improvement relative to the preceding configuration.

From Figure 11, it is evident that pose normalization and the AFP module contribute the most to performance, with particularly notable gains on AffectNet and ExpW. This further confirms the effectiveness of multi-stage feature optimization and pose correction. From Table 1, pose normalization uses Qwen-image-edit [31] to frontalize pose-abnormal samples (17.5% in RAF-DB, 23.8% in AffectNet-7, and 19.2% in EXPW), yielding an average accuracy improvement of 1.30% across the three datasets, with the largest gain on AffectNet-7 (+1.45%) due to more in-the-wild pose variations. The robust standardization strategy based on the median and interquartile range (IQR) improves accuracy by an average of 0.92% over the three datasets, with the most significant contribution on RAF-DB (+1.37%), showing stronger robustness to outliers from illumination changes and noise. PCA dimensionality reduction from the original 768 dimensions to 512 (retaining 98.1% variance) brings an average accuracy increase of 0.39% and shortens classifier training time by about 40%; the effect is slightly more pronounced on the large-scale in-the-wild EXPW dataset (+0.43%). The class-balancing strategy, which assigns higher weights to minority classes (e.g., disgust and fear), improves average accuracy by 0.30% across the three datasets and effectively boosts minority recall: on RAF-DB, the disgust F1 increases from 0.64 to 0.67 (+4.7%), and on AffectNet-7, the fear F1 increases from 0.51 to 0.54 (+5.9%). The soft-voting ensemble of five logistic regressions (LR) and one Support Vector Machine (SVM) yields an average accuracy gain of 0.43% across the three datasets, outperforming hard voting by 0.32 percentage points, with the best effect on RAF-DB (+0.66%)—a medium-to-small-scale dataset. Cross-dataset consistency analysis shows cumulative accuracy gains of +4.19% (RAF-DB), +2.95% (AffectNet-7), and +2.89% (EXPW) for the full model, averaging a 3.34-point improvement. The consistent contributions of each component validate the proposed method’s generalization and robustness.

#### 4.5.3. Comparison of Feature Extractors

To validate the advantages of CLIP pre-trained features over traditional vision models, we compare four feature extractors on RAF-DB, AffectNet-7, and EXPW: ResNet-50 (ImageNet pre-trained), ViT-B/16 (ImageNet-21K pre-trained), CLIP ViT-B/32, and CLIP ViT-L/14 (LAION-2B pre-trained). The results are shown in Table 5.

From the table, CLIP ViT-L/14’s average accuracy (72.25%) is 3.52 points higher than ResNet-50 and 2.28 points higher than ViT-B/16. This stems from its image–text contrastive learning paradigm: trained on 2B image–text pairs, it learns multimodal semantic representations that more precisely capture abstract emotional concepts in facial expressions. Even within the ViT family, CLIP ViT-L/14 pre-trained on 2B samples significantly outperforms ViT-B/16 pre-trained on 14M samples, improving average accuracy by 2.28 points and confirming the decisive impact of large-scale pre-training on transfer performance. Unlike ResNet and ViT that learn solely in the visual domain, CLIP aligns visual features with language semantics by matching images with descriptive text, reducing the difficulty of feature-to-label mapping in downstream tasks and improving robustness to ambiguous or deformed expressions in the wild. Meanwhile, CLIP ViT-L/14 (768-d features) averages 1.12 points higher accuracy than CLIP ViT-B/32 (512-d features); the larger capacity captures more expression detail while maintaining a good balance of performance and computational efficiency. Therefore, we select CLIP ViT-L/14 as the feature extractor.

#### 4.5.4. Sensitivity Analysis of Parameter Settings

To identify the best balance between information retention and overfitting prevention in PCA, we evaluate different PCA dimensions (128, 256, 384, 512, 640, and 768) on RAF-DB. As shown in Table 6, 512 dimensions is optimal; it retains 98.1% of the original variance and achieves the highest accuracy (88.32%). When the dimension falls below 384, accelerated information loss causes a marked accuracy drop; above 512, redundant information induces mild overfitting. Moreover, the 512-d setting shortens training time from 45.2 s (original 768-d) to 25.3 s (about 44% reduction), striking a balance among information retention, performance, and efficiency.

To evaluate how different ensembling strategies affect performance, we compare five strategies across three datasets. As shown in Table 7, a single LR achieves a markedly higher average accuracy than a single RF (+3.17% on average). Soft voting, which fuses classifier output probabilities, outperforms hard voting by an average of 0.32 points. The soft-voting ensemble of 5 LRs plus 1 SVM achieves the best performance by leveraging the complementarity between the SVM and linear LRs to increase classifier diversity, improving average accuracy by a further 0.06% over LR+RF soft voting. Therefore, the 5 × LR + 1 × SVM soft-voting ensemble is our optimal configuration.

#### 4.5.5. Effectiveness of Pose Normalization

To verify the improvement brought by Qwen-image-edit pose normalization on profile-face expression recognition, we evaluate accuracy before and after normalization on profile subsets (yaw > 30°) for the three datasets.As shown in Table 8.

Pose normalization increases profile-sample accuracy by an average of 6.72 points across the three datasets, with highly consistent gains (6.61–6.82%). It effectively corrects geometric distortions in profile faces and restores completeness of key expression regions. Before normalization, the average accuracy on profile samples (58.22%) is notably lower than the overall accuracy (70%) by about 12 points, due to geometric deformation, uneven lighting, and distribution shift. Compared with traditional 3D reconstruction methods (e.g., 3DDFA), Qwen-image-edit—pre-trained for large-scale image editing—better understands facial pose structure and converts profiles to frontal views more naturally while preserving emotional consistency, and is much faster (0.8 s/image vs 5.2 s/image for 3DDFA). After normalization, the average accuracy on profile samples (64.94%) remains slightly below the overall accuracy, with the gap reduced to about 5 points, indicating room for further improvement, potentially via specialized profile-expression training data or multi-view feature fusion.

Figure 12 compares attention heatmaps on EXPW and RAF-DB before and after normalization: post-normalization, attention for all emotion classes mostly concentrates on the face. Improvements are most pronounced for easily confused classes such as disgust (+8.3%) and fear (+9.1%), further validating the benefits of pose normalization. Moreover, the Qwen-image-edit model—purpose-built for image editing—adjusts facial pose while better preserving the integrity and consistency of emotional expression, avoiding attention distortion.

Figure 13 compiles representative failure cases, including (1) extreme backlighting causing PFN to oversmooth facial shadows; and (2) mild identity drift in cluttered backgrounds. Coupled with the quality-control workflow in Section 4.3, we compared performance before and after filtering: using every frontalized image directly reduces overall accuracy on RAF-DB, AffectNet-7, and ExpW by 0.42–0.58 percentage points. By enforcing joint thresholds on CLIP semantic similarity (0.82) and ArcFace identity similarity (0.78), followed by manual review, we revert 1.9%, 2.1%, and 2.4% of the samples to their original images, restoring accuracy to the level reported in the main text.

#### 4.5.6. Cross-Dataset Generalization Evaluation

We train QC-FER on RAF-DB and directly test it on the side-face subsets of AffectNet-7 and ExpW. Without fine-tuning, the accuracies reach 56.1% and 58.7%, exceeding an end-to-end ResNet-50 baseline by 3.5 and 4.1 percentage points. Removing PFN while retaining CLIP + AFP + ESV reduces the accuracy to 53.2% and 55.4%, confirming that pose normalization is equally critical for cross-domain adaptation. Because CK+, 3DFE, and other indoor datasets are not yet covered, this evaluation still has room to expand.

#### 4.5.7. Complexity and Efficiency Analysis

Including CLIP feature extraction adds a shared preprocessing cost of 12.4 ms per image and 7.2 GB of peak memory for all methods. Compared with end-to-end networks, as show in Table 9, QC-FER reduces training time by roughly five-fold and cuts the trainable parameter count to 1/27. Because inference consists only of matrix operations, latency and memory usage are markedly lower than those of deep models that require full convolutional forward passes.

#### 4.5.8. Limitation

As shown in Figure 14, the qwen-image-edit model can introduce facial component distortions due to erroneous generation, such as changes in mouth openness and eye aperture. This indicates that although QC-FER significantly improves robustness in complex environments, it may still produce local feature errors under extreme conditions. To comprehensively assess performance across emotion classes, we generated confusion matrices on three FER datasets (RAF-DB, AffectNet-7, and EXPW), as shown in Figure 15. The confusion matrices reveal several key patterns. All three datasets exhibit a clear diagonal dominance, indicating good overall discriminability. “Happy” achieves the highest recall across all datasets (RAF-DB: 95%, AffectNet-7: 89%, and EXPW: 82%), benefiting from its distinctive smiling feature (AU12). Major confusion patterns include “Disgust vs. Anger” (19% on RAF-DB, 21% on AffectNet-7), both sharing negative valence and tense facial muscles (frowning, tight lips); “Fear vs. Surprise” (15% on RAF-DB), as both display wide-open eyes and raised eyebrows (AU1, AU2, and AU5), differing mainly in mouth shape. “Disgust” has the highest error rates across datasets (RAF-DB: 37%, AffectNet-7: 49%, EXPW: 45%), attributable to fewer samples (only 108 in RAF-DB, 3.9%) and feature ambiguity. Despite class-weighting, further improvement is needed under severe imbalance.

We have only explored freezing CLIP so far; fine-tuning under limited data leads to overfitting. Scalable parameter-efficient adaptation remains an open direction.

## 5. Conclusions

QC-FER integrates PFN pose normalization, CLIP semantic embeddings, AFP adaptive preprocessing, and ESV ensemble classification to enhance facial expression recognition under extreme poses. It achieves accuracies of 89.39%, 62.66%, and 67.17% on RAF-DB, AffectNet-7, and ExpW, respectively—an average improvement of 3.34 percentage points over the baselines. On profile subsets, the average gain reaches 6.72 points, validating PFN’s ability to correct geometric distortions. Module-level and component-level ablations further demonstrate complementary contributions, and the shallow ensemble attains performance comparable to or better than deeper networks when operating on frozen CLIP embeddings.

## Figures and Tables

**Figure 1 jimaging-12-00024-f001:**
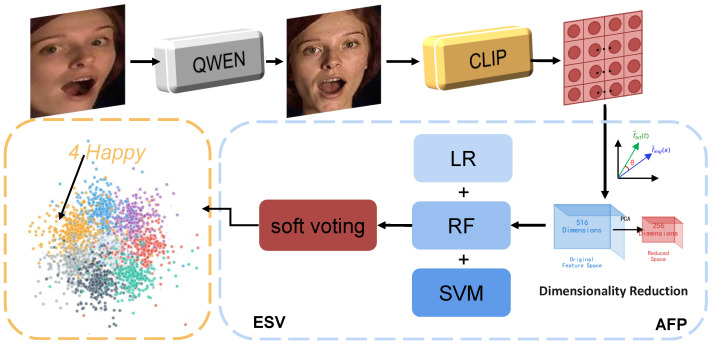
Outline of the proposed QC-FER. QC-FER identifies the emotional state of the target through four components. Qwen denotes qwen-edit-image; CLIP refers to CLIP ViT-L/14; AFP stands for the Adaptive Feature Preprocessing module; ESV represents the Ensemble Soft Voting classification module. LR indicates logistic regression, RF denotes Random Forest, and SVM refers to Support Vector Machine.

**Figure 2 jimaging-12-00024-f002:**
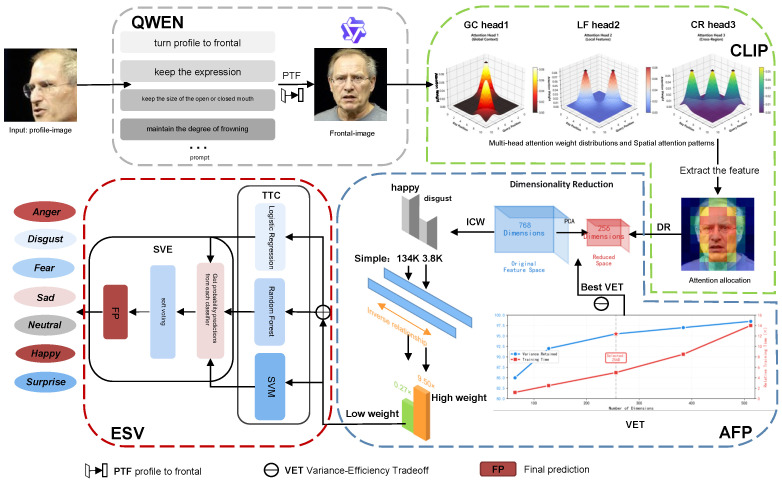
Overview of our QC-FER framework for in-the-wild facial expression recognition (FER). The input image is optimized by QWEN, then CLIP extracts features. AFP reduces feature dimensionality and assigns weights to training data with long-tailed distribution. ESV ensembles multiple base learners to perform emotion classification via soft voting.

**Figure 3 jimaging-12-00024-f003:**
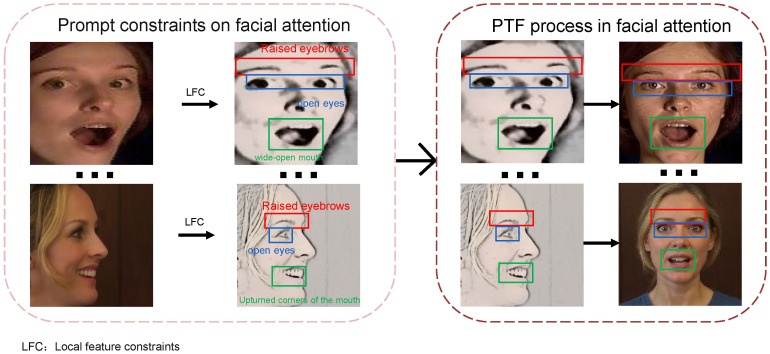
Prompt constraints and generation process.

**Figure 4 jimaging-12-00024-f004:**
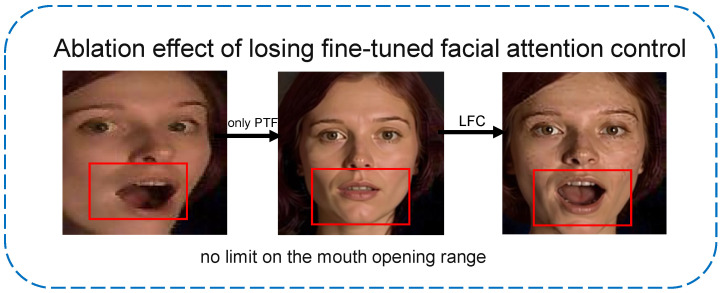
Effects of missing local feature constraints.

**Figure 5 jimaging-12-00024-f005:**
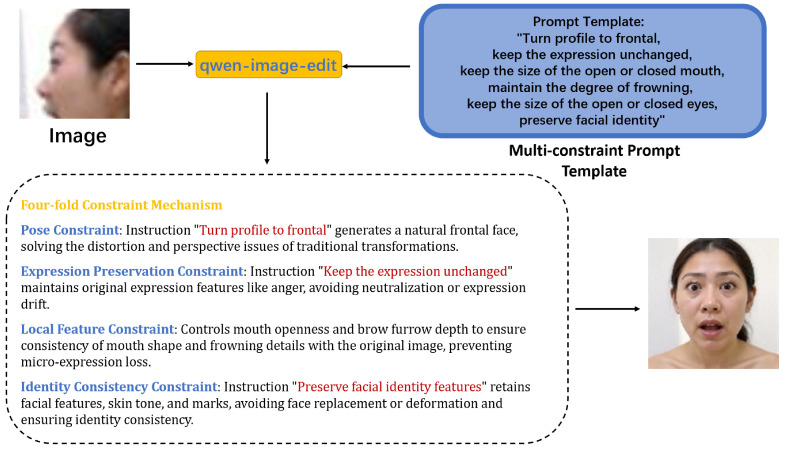
Illustration of multi-constraint prompt design for face normalization.

**Figure 6 jimaging-12-00024-f006:**
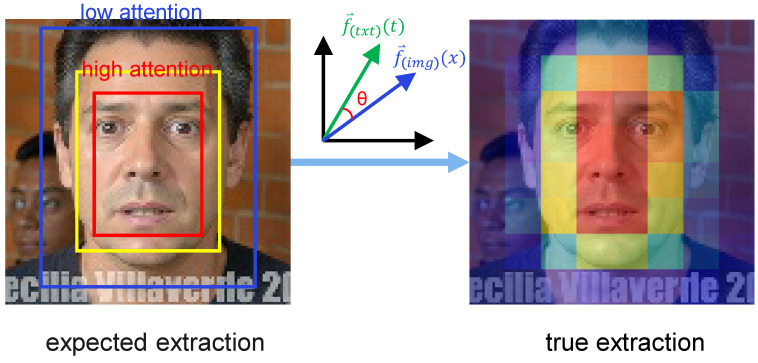
Illustration of attention differences between desired and actual extraction in CLIP feature extraction.

**Figure 7 jimaging-12-00024-f007:**
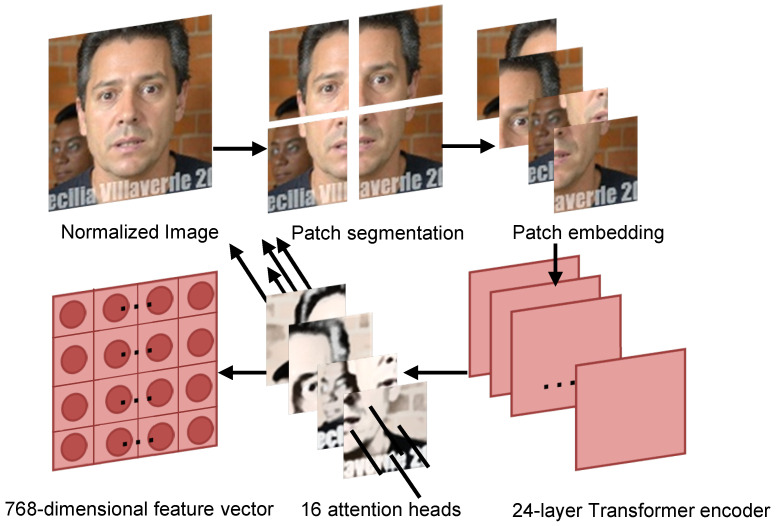
Illustration of the patch segmentation, embedding, and Transformer encoder processing workflow.

**Figure 8 jimaging-12-00024-f008:**
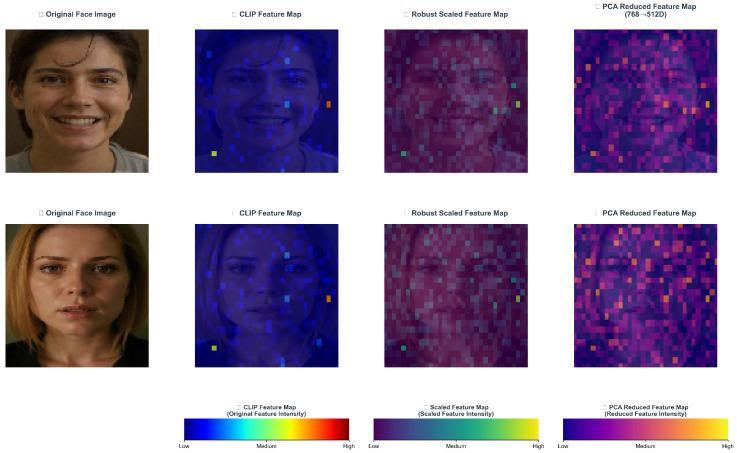
Visualization of AFP feature preprocessing results.

**Figure 9 jimaging-12-00024-f009:**
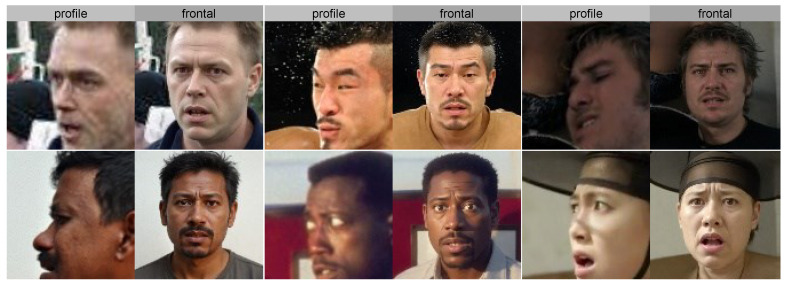
Profile → frontal conversion examples (diverse samples).

**Figure 10 jimaging-12-00024-f010:**
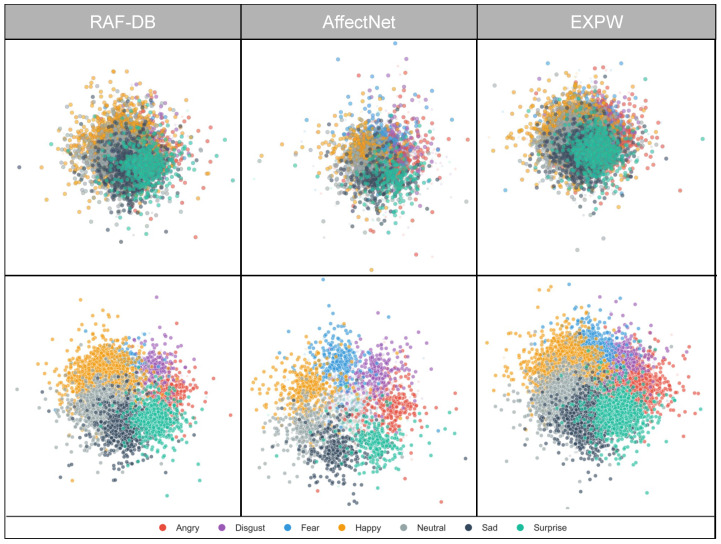
SNE feature visualization (comparison: top row = baseline; bottom row = after QC-FER processing).

**Figure 11 jimaging-12-00024-f011:**
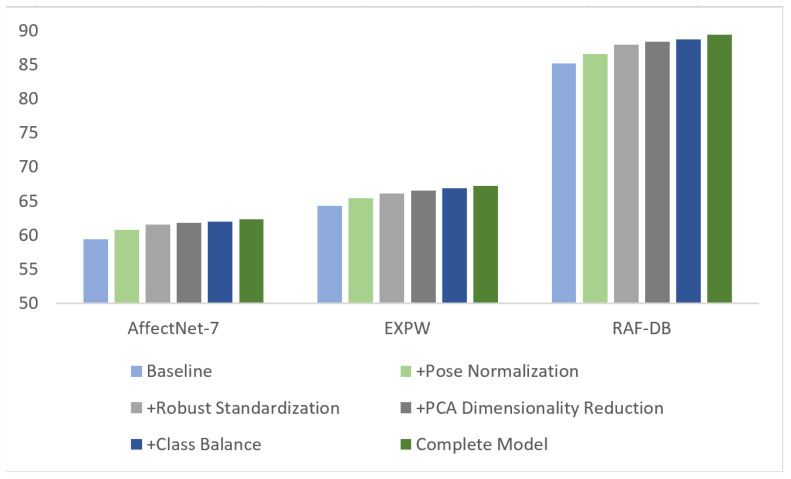
Comparative bar chart of module contribution on three datasets.

**Figure 12 jimaging-12-00024-f012:**
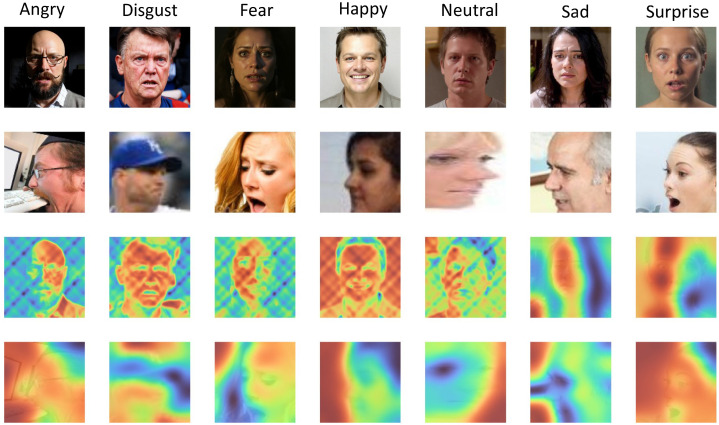
Activation map visualization on different emotion categories.

**Figure 13 jimaging-12-00024-f013:**
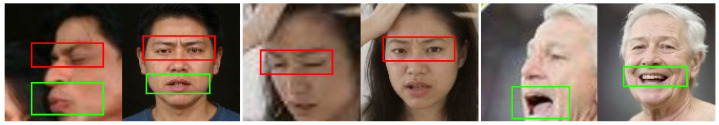
Key regions of challenging samples.

**Figure 14 jimaging-12-00024-f014:**
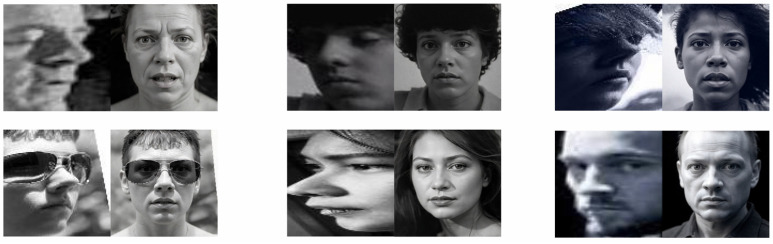
Visualization of difficult samples and failure cases.

**Figure 15 jimaging-12-00024-f015:**
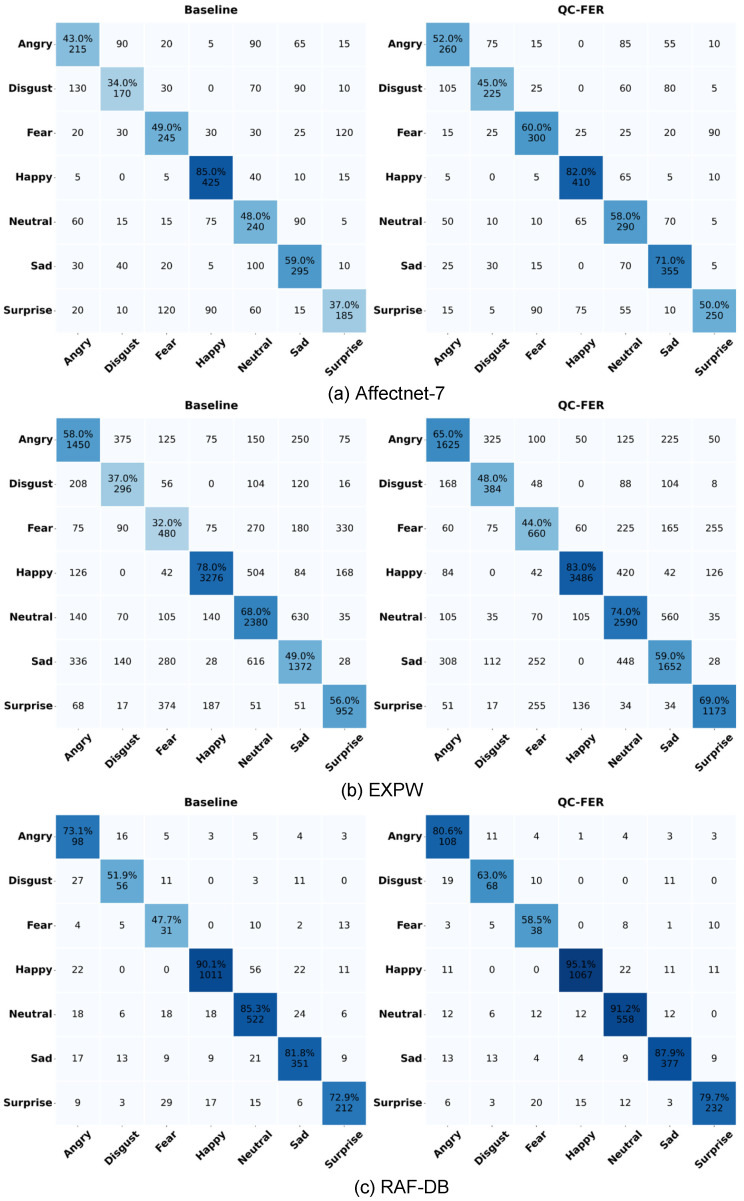
Recognition confusion matrices on three FER in-the-wild datasets by baseline and the proposed QC-FER. (**a**) AffectNet-7; (**b**) RAF-DB; (**c**) SFEW.

**Table 1 jimaging-12-00024-t001:** Performance comparison of QC-FER with state-of-the-art methods on three FER datasets.

Database	Method	Accuracy	Database	Method	Accuracy	Database	Method	Accuracy
>RAF-DB	EfficientFace [36]	88.36%	AffectNet	IPA2LT [37]	55.11%	EXPW	CADA [38]	59.40%
	MA-Net [23]	88.40%		gACNN [39]	58.78%		ICID [40]	60.04%
	PAT-ResNet-101 [15]	88.43%		SPWFA_SE [1]	59.23%		SWD [16]	60.64%
	WSFER [41]	88.89%		RAN [42]	59.50%		HOG+SVM [43]	60.66%
	RUL [13]	88.98%		SCN [44]	60.23%		SAFN [45]	61.40%
	SPLDL [46]	89.08%		FENN [37]	60.83%		JUMBOT [23]	63.69%
	TAN [18]	89.12%		DENet [12]	60.94%		RANDA [5]	63.87%
	HistNet [29]	89.24%		RUL [13]	61.43%		AGRA [16]	63.94%
	AMP-Net [32]	89.25%		MTAC [35]	61.58%		CGLRL [28]	64.87%
	ADDL [22]	89.34%		EASE [34]	61.82%		Baseline DCN [43]	65.06%
	DMUE [11]	89.42%		MRAN [33]	62.48%		LPL [39]	66.90%
	QC-FER (our)	89.39%		QC-FER (our)	62.66%		QC-FER (our)	67.17%

**Table 2 jimaging-12-00024-t002:** A summary of the per-class precision, recall, and F1-score across the three datasets.

Emotion	RAF-DB P	RAF-DB R	RAF-DB F1	AffectNet-7 P	AffectNet-7 R	AffectNet-7 F1	ExpW P	ExpW R	ExpW F1
Angry	80.6	86.7	83.5	64.2	52.0	57.5	65.0	66.4	65.7
Disgust	63.0	74.2	68.1	48.6	45.0	46.7	48.0	52.7	50.2
Fear	58.5	83.1	68.7	58.3	60.0	59.1	44.0	59.3	50.5
Happy	95.1	95.2	95.2	71.0	82.0	76.1	83.0	76.9	79.8
Neutral	91.2	89.5	90.3	64.8	58.0	61.2	74.0	64.8	69.1
Sad	87.9	86.1	87.0	60.4	71.0	65.3	59.0	60.1	59.5
Surprise	79.7	91.7	85.3	67.5	50.0	57.4	69.0	68.5	68.8

**Table 3 jimaging-12-00024-t003:** Performance comparison of different module configurations.

Configuration	PFN	AFP	ESV	RAF-DB	AffectNet-7	ExpW
CLIP Baseline	No	No	No	85.20	59.34	64.28
PFN + CLIP	Yes	No	No	86.51	60.79	65.42
PFN + CLIP + AFP	Yes	Yes	No	88.73	61.97	66.85
QC-FER (Full Pipeline)	Yes	Yes	Yes	89.39	62.29	67.17

**Table 4 jimaging-12-00024-t004:** Ablation results of core components (accuracy, %).

Configuration	Pose Norm	Robust Std	PCA DimRed	Class Balance	Ensemble	RAF-DB	AffectNet-7	EXPW
Baseline	✗	✗	✗	✗	✗	85.20	59.34	64.28
+Pose Normalization	✓	✗	✗	✗	✗	86.51 (+1.31)	60.79 (+1.45)	65.42 (+1.14)
+Robust Standardization	✓	✓	✗	✗	✗	87.88 (+1.37)	61.53 (+0.74)	66.08 (+0.66)
+PCA Dimensionality Reduction	✓	✓	✓	✗	✗	88.32 (+0.44)	61.82 (+0.29)	66.51 (+0.43)
+Class Balance	✓	✓	✓	✓	✗	88.73 (+0.41)	61.97 (+0.15)	66.85 (+0.34)
Complete Model	✓	✓	✓	✓	✓	89.39 (+0.66)	62.66 (+0.59)	67.17 (+0.32)

**Table 5 jimaging-12-00024-t005:** Performance comparison of different feature extractors on three datasets (accuracy, %).

Feature Extractor	Pre-training Data	Feature Dim	RAF-DB	AffectNet-7	EXPW
ResNet-50 (ImageNet)	1.2 M images	2048	84.30	58.70	63.20
ViT-B/16 (ImageNet-21K)	14 M images	768	85.60	59.80	64.50
CLIP ViT-B/32	400 M image–text pairs	512	87.59	61.20	65.80
CLIP ViT-L/14 (LAION-2B)	2 B image–text pairs	768	89.39	62.66	67.17

**Table 6 jimaging-12-00024-t006:** Effect of PCA dimensions on performance (RAF-DB).

PCA Dim.	Explained Variance (%)	Accuracy (%)	Training Time (s)
128	83.7	84.71	8.2
256	91.2	86.34	12.5
384	95.8	87.02	18.7
512	98.1	88.32	25.3
640	99.5	88.17	34.8
768	100.0	87.95	45.2

**Table 7 jimaging-12-00024-t007:** Performance comparison of different ensembling strategies (accuracy, %).

Ensemble Strategy	RAF-DB	AffectNet-7	EXPW
Single LR	88.32	62.29	66.85
Single RF	85.86	58.42	63.18
Hard Voting (LR + RF)	88.85	62.05	66.92
Soft Voting (LR + RF)	89.12	62.41	67.08
Completed Soft Voting	89.39	62.66	67.17

**Table 8 jimaging-12-00024-t008:** Impact of pose normalization on profile-face recognition accuracy.

Dataset	Profile Samples	Pre-Norm Acc (%)	Post-Norm Acc (%)	Improvement
AffectNet-7	121	52.07	58.68	+6.61
RAF-DB	308	74.35	81.17	+6.82
EXPW	1878	48.23	54.96	+6.73

**Table 9 jimaging-12-00024-t009:** Comparison of QC-FER with end-to-end deep FER models under identical hardware (RTX 3050, 8 GB). All methods use offline CLIP ViT-L/14 features; trainable parameters count only components updated via gradient descent. DMUE trains end-to-end; MA-Net includes transformer attention; 3DDFA-V2 + FFGAN incorporates 3D fitting and GAN-based reconstruction; QC-FER trains only lightweight LR/SVM/RF classifiers with cached CLIP features.

Method	Trainable Parameters	Per-Image Inference Time (ms)	Training Time (per Epoch)	Peak Memory
DMUE (ResNet-50)	43.2 M	16.3	21.4 h	10.8 GB
MA-Net (ResNet-101)	55.7 M	18.9	24.7 h	12.1 GB
3DDFA-V2 + FFGAN	18.6 M	21.5	26.0 h	9.6 GB
QC-FER (ours)	1.6 M	6.8	4.2 h	3.1 GB

## Data Availability

The data presented in this study are openly available in GitHub at https://github.com/users/1574087287cbh-ux/projects/2/views/1 (accessed on 25 December 2025).

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
