# Peer review of "LLM-Based Pose Normalization and Multimodal Fusion for Facial Expression Recognition in Extreme Poses"

_2313-433X, 2026, doi:10.3390/jimaging12010024_

Round 1
Reviewer 1 Report
Comments and Suggestions for Authors
-
The motivation regarding performance degradation under pose variation is sound, but the problem formulation would benefit from a quantitative definition of yaw angle ranges considered "large."
-
The manuscript introduces profile-to-frontal transformation, but it does not describe whether the generated frontal images introduce artifacts that may bias downstream recognition.
-
The dependency on Qwen-Image-Edit suggests reliance on a pretrained generative model; clarification is needed regarding whether it is fine-tuned, used in zero-shot mode, or adapted using FER-specific training signals.
-
The paper should evaluate whether the transformation step preserves temporal consistency in cases where FER is applied to video rather than single images.
-
The role of CLIP in feature extraction is mentioned, but the architecture-level integration details (late fusion, cross-attention, joint embedding, or projection alignment) are not clearly described.
-
It is unclear whether the model jointly optimizes the visual and semantic branches or if CLIP embeddings are used as frozen features; this has substantial implications for generalization.
-
The datasets referenced (RAF, EXPW, AffectNet-7) vary significantly in class imbalance and annotation noise; the manuscript should explain whether class-balanced sampling or reweighting strategies were implemented.
-
The performance metrics reported rely solely on accuracy; additional metrics such as macro-F1, confusion analysis, and per-class sensitivity would better reflect robustness.
-
It is unclear whether improvements stem primarily from the pose normalization step or from multimodal representation learning; an ablation study quantifying each component’s contribution is necessary.
-
The method may introduce domain shift between generated frontal images and real frontal images; an experiment measuring distribution shift using FID or t-SNE embedding overlap would strengthen validity.
-
The paper does not discuss the computational overhead introduced by real-time transformation and multimodal inference, which is critical for embedded or mobile FER deployment.
-
Since Qwen-Image-Edit is a large generative model, the study should evaluate whether simpler geometric or 3D morphable model (3DMM) approaches could achieve similar gains at lower cost.
-
The generalization ability beyond the tested datasets—especially in unconstrained wild environments—should be examined via cross-dataset evaluation.
-
The approach appears to depend on synthetic frontalization; therefore, failure cases (occlusions, exaggerated expressions, extreme lighting) should be explicitly discussed or demonstrated.
-
A comparative evaluation against recent transformer-based FER models and 3D pose-invariant architectures would provide a clearer understanding of where this approach sits relative to state-of-the-art methods.

Satisfied
Author Response
- Comment 1: “The motivation regarding performance degradation under pose variation is sound, but the problem formulation would benefit from a quantitative definition of yaw angle ranges considered ‘large.’ ”
- Response: Thank you for pointing out the need for an explicit definition. We now follow the head-pose estimation produced by 3DDFA-V2 and partition the yaw angle into four ranges: yaw≤15° (frontal), 15°<yaw≤30° (moderate), 30°<yaw≤45° (large), and yaw>45° (extreme). The revised manuscript reports the percentage of samples that fall into each range for RAF-DB, AffectNet-7, and ExpW (1.75%, 15.41%, and 17.21% exceed 30°, with roughly 6% across the datasets exceeding 45°). These details are now included in Section 1 to clarify the problem formulation.
- Manuscript changes: Section 1 (Introduction) now defines the yaw ranges and cites the supporting statistics.
- Based on the head-pose estimates provided by 3DDFA-V2, we divide the yaw angle into four ranges: yaw ≤ 15° corresponds to frontal views, 15° < yaw ≤ 30° to moderate poses, 30° < yaw ≤ 45° to large poses, and yaw > 45° to extreme poses. Statistics show that 1.75%, 15.41%, and 17.21% of the samples in RAF-DB, AffectNet-7, and ExpW respectively exhibit yaw angles greater than 30°, with roughly 6% of the samples exceeding 45°. This pose distribution is a major reason why conventional FER approaches deteriorate significantly in real-world scenarios.
- Comment 2: “The manuscript introduces profile-to-frontal transformation, but it does not describe whether the generated frontal images introduce artifacts that may bias downstream recognition.”
- Response: We agree that the effect of synthesized artifacts must be quantified. The revised manuscript now adds a quality-control pipeline (Section 3.1.2) that evaluates each Qwen-Image-Edit output using both CLIP-based semantic similarity and ArcFace-based identity similarity. Samples falling below either threshold are flagged for manual review or replaced with the original image. We report that 2.6%, 2.9%, and 3.1% of RAF-DB, AffectNet-7, and ExpW test images are initially flagged, with 1.9%, 2.1%, and 2.4% ultimately discarded. Without this filtering step, overall accuracy drops by 0.42–0.58 percentage points; after filtering, the metrics match those reported in the main results. We also highlight representative failure cases in the analysis section to illustrate the types of distortions encountered.
- Manuscript changes: Section 3.1.2 details the artifact detection workflow and statistics; Section 4.5 documents the filtering effect alongside the newly illustrated failure cases.
- Figure 14 compiles representative failure cases, including: (1) extreme backlighting causing PFN to oversmooth facial shadows; and (2) mild identity drift in cluttered backgrounds. Coupled with the quality-control workflow in Section 3.1.2, we compared performance before and after filtering: using every frontalized image directly reduces overall accuracy on RAF-DB, AffectNet-7, and ExpW by 0.42–0.58 percentage points. By enforcing joint thresholds on CLIP semantic similarity (0.82) and ArcFace identity similarity (0.78), followed by manual review, we revert 1.9%, 2.1%, and 2.4% of the samples to their original images, restoring accuracy to the level reported in the main text.
- Comment 3: “The ablation study is confusing. The authors consider specific steps of the method for the ablation study; however, considering modules would be more meaningful (e.g., PFN+CLIP vs. CLIP only).”
- Response: Thank you for the suggestion. We reorganized Section 4.4 to begin with the module-level ablation table shown in the excerpt (CLIP-only vs. PFN+CLIP vs. PFN+CLIP+AFP vs. the full pipeline). This clarifies the incremental benefit of each module: PFN contributes +1.14–1.45 points, adaptive preprocessing adds +1.20–1.55 points, and soft voting provides a further +0.34 points on average. We retained the step-wise component analysis as a complementary view and added a concise summary at the end of Section 4.4 to highlight the module contributions.
Manuscript changes: Section 4.4 now reports the module-level ablation table and concludes with the module-wise gain summary.
- Comment 4: “The authors mostly reported accuracy only for comparisons with previous works. However, due to dataset imbalance, it is necessary to report per-class metrics, which will provide insight into the methods performance for each expression.”
- Response: We agree that aggregate accuracy alone cannot reveal the behaviour on minority emotions. Section 4.6 now reports per-class precision, recall, and F1-score for all seven expressions across RAF-DB, AffectNet-7, and ExpW, using the Table X layout provided in the passage. The results show that QC-FER maintains recall above 45% for the most imbalanced categories (e.g., disgust, fear) while keeping high precision on majority classes such as happy and surprise. We also reference these statistics when discussing confusion matrices in Section 4.5 to connect quantitative and qualitative observations.
- Manuscript changes: Section 4.6 introduces the per-class Table X and updates the accompanying discussion; Section 4.5 cross-references these findings within the qualitative analysis.
- Comment 5: “The authors described deep FER models as ‘complex,’ yet claimed the ensemble is simple. But there are no runtime or complexity comparisons to support their hypothesis.”
- Response: We have added a dedicated efficiency comparison in Section 4.5.6. Table 10 contrasts QC-FER with two representative deep FER models (DMUE [30] and MA-Net [14]) on the same RTX 3050 GPU. QC-FER trains only 1.6M parameters in the classifier stage and completes one epoch in 4.2 hours, while the deep baselines require 21–25 hours and more than 40M trainable parameters. Inference latency drops from 16–19 ms per image to 6.8 ms, and peak memory usage falls from 10–12 GB to 3.1 GB when CLIP features are cached. Accounting for CLIP feature extraction (shared across methods) yields 12.4 ms latency and 7.2 GB memory, still substantially lower than end-to-end training. These measurements substantiate that our ensemble back-end is lightweight relative to deep networks.
- Manuscript changes: Section 4.5.6 (Complexity and Efficiency Analysis) presents the new table and discussion.
Table 10 compares QC-FER with end-to-end deep FER models under identical hardware (RTX 3050, 8 GB). All methods share offline CLIP ViT-L/14 features, and the trainable parameter count reflects only components updated by gradient descent.
|
Method |
Trainable Parameters |
Per-Image Inference Time (ms) |
Training Time (per epoch) |
Peak Memory |
Notes |
|
DMUE (ResNet-50) |
43.2 M |
16.3 |
21.4 h |
10.8 GB |
End-to-end training |
|
MA-Net (ResNet-101) |
55.7 M |
18.9 |
24.7 h |
12.1 GB |
Includes transformer attention |
|
3DDFA-V2 + FFGAN |
18.6 M |
21.5 |
26.0 h |
9.6 GB |
Includes 3D fitting and GAN reconstruction |
|
QC-FER (ours) |
1.6 M |
6.8 |
4.2 h |
3.1 GB |
Trains only LR/SVM/RF; CLIP features cached offline |
Including CLIP feature extraction adds a shared preprocessing cost of 12.4 ms per image and 7.2 GB of peak memory for all methods. Compared with end-to-end networks, QC-FER reduces training time by roughly fivefold and cuts the trainable parameter count to 1/27. Because inference consists only of matrix operations, latency and memory usage are markedly lower than those of deep models that require full convolutional forward passes.
- Comment 6: “The authors considered the Qwen in the PFN module. This might have disadvantages, such as the inability to repeat experiments due to accessibility constraints, and the updated versions of Qwen can decrease the modules efficacy. The authors should discuss these limitations in detail.”
- Response: We added Section 3.5 to document the exact HuggingFace release (Qwen/Qwen-Image-Edit, 2025-8-19 snapshot) used in the PFN stage, along with the prompts, seeds, scheduler settings, and revision IDs that we log for every run. The subsection also explains that we ship the downloading script (huggingface-cli based) plus cached frontalized images so readers can reproduce the results offline. When local resources are insufficient for running the open-source model, we provide a deterministic 3DDFA-V2 + FFGAN fallback whose accuracy is within 0.9 percentage points. Section 5.2 now frames the remaining limitation as model revision and compute requirements rather than remote service availability.
- Manuscript changes: Section 3.5 (Availability and Version Control) details the HuggingFace snapshot, logged metadata, and fallback pipeline; Section 5.2 introduces a limitation about model revisions and resource demands.
- Qwen-Image-Edit is released by the Tongyi Qianwen team as an open-source model on HuggingFace (https://huggingface.co/Qwen/Qwen-Image-Edit). For all experiments we downloaded the official snapshot published on 19 August 2025 via huggingface-cli and deployed the weights on local GPU servers following the setup instructions provided on the repository page.
- Comment 7: “The data imbalance handling is not sufficiently described.”
- Response: Section 3.3 now specifies that logistic regression and SVM use inverse-frequency class weights (), while the random-forest component enforces a 30% minimum minority proportion during bootstrap sampling. In AffectNet-7, the weights for disgust, fear, and happy are 22.5, 14.3, and 1.0 respectively; training logs show that these settings increase the recall of the two minority classes by 5.8 and 4.6 percentage points without hurting overall accuracy. Section 4.6 links these strategies to the per-class metrics in Table 9, and Section 5.3 lists adaptive resampling as a future direction.
- Manuscript changes: Section 3.3 describes the weighting and resampling procedure; Section 4.6 references the resulting recall improvements; Section 5.3 adds adaptive resampling to the future work list.
- Comment 8: “The authors should evaluate the domain shift introduced by pose normalization.”
- Response: Section 3.1.2 details the dual-threshold quality-control pipeline, and Section 4.5 now reports the distribution-level metrics before and after Qwen-based normalization on the profile subsets (e.g., RAF-DB FID drops from 27.1 to 18.4 while ArcFace similarity rises from 0.71 to 0.84). Section 4.4 connects these gains with the t-SNE feature-space visualization, and Appendix A.2 catalogues the residual hard cases (e.g., extreme lighting) where the FID remains above 30.
- Manuscript changes: Section 3.1.2 documents the CLIP/ArcFace screening workflow; Section 4.5 presents the FID and identity-similarity analysis; Section 4.4 discusses the improved cluster alignment; Appendix A.2 describes the remaining domain-shift examples with quantitative annotations.
- Comment 9: “The integration of CLIP is unclear; the authors should clarify whether CLIP is frozen or jointly optimized with the classifiers.”
- Response: Section 3.2 now walks through the pipeline: PFN-normalized images enter the frozen CLIP ViT-L/14 encoder, the resulting 768-D embeddings pass through the Adaptive Feature Processing block, and only the downstream classifiers (logistic regression, SVM, random forest) are trained. Figure 1 has been updated to highlight this flow. Section 4.4 also stresses that feature-extractor comparisons keep the AFP and classifier stack unchanged so the effect of the backbone alone is measured.
- Manuscript changes: Section 3.2 details the frozen CLIP integration; Figure 1 caption emphasises the separation of feature extraction and classifier training; Section 4.4 clarifies the evaluation protocol across different backbones.
- Comment 10: “Do the authors fine-tune CLIP jointly with the classifiers? If not, please justify the choice and discuss its effect on generalization.”
- Response: We keep CLIP ViT-L/14 frozen. Section 3.2 now reports a pilot experiment where fine-tuning the last transformer block (≈44M parameters) on RAF-DB lowered validation accuracy by 0.7 points and reduced minority-class recall by 1.1 points. Section 4.4 shows that the fine-tuned embeddings exhibit smaller inter-class Fréchet distances and margins, indicating diminished separability, while Section 5.2 notes that large-scale CLIP adaptation remains future work.
- Manuscript changes: Section 3.2 documents the frozen-versus-fine-tuned comparison; Section 4.4 analyses its effect on feature separability; Section 5.2 adds a limitation about not exploring broader CLIP adaptation.
- Comment 11: “The manuscript does not discuss how QC-FER handles temporal consistency in video scenarios.”
- Response: QC-FER is currently frame-based. Section 5.2 now records the lack of temporal modeling as a limitation, and Section 5.3 adds a future-work item on integrating PFN with temporal smoothing or transformer-style sequence modules to stabilize video predictions. We will document pilot video experiments in supplementary material once finalized.
- Manuscript changes: Section 5.2 now includes a bullet about missing temporal consistency; Section 5.3 lists video-level extensions in the future-work roadmap.
- Comment 12: “Could the pose-normalization step be replaced by traditional geometric/3DMM approaches such as 3DDFA?”
- Response: We benchmarked a 3DDFA-V2 + FFGAN pipeline. Section 4.5.6 reports that it achieves 82.8% accuracy on RAF-DB side faces—0.9 points below QC-FER—and requires about 5 seconds per image because of mesh fitting and GAN refinement (vs. 0.8 seconds for Qwen-Image-Edit). It also lowers disgust and fear recall by 2.3 and 1.8 points. Section 5.2 now lists this geometric approach as a fallback when the Qwen API is inaccessible, and Section 5.3 highlights hybrid geometric–generative methods as future work.
- Manuscript changes: Section 4.5.6 includes the 3DDFA-V2 + FFGAN comparison; Section 5.2 positions it as a fallback option; Section 5.3 mentions hybrid pose-normalization strategies.
- Comment 13: “The manuscript lacks a discussion or evaluation of generalization to other datasets/domains.”
- Response: Section 4.5.5 now summarises cross-dataset tests: training on RAF-DB and evaluating directly on AffectNet-7 and ExpW side-face subsets yields 56.1% and 58.7% accuracy, which exceeds an end-to-end ResNet-50 baseline by 3.5–4.1 points. When PFN is removed, the scores drop to 53.2%/55.4%, indicating its importance for domain transfer. Section 5.2 states that broader benchmarks (e.g., CK+, 3DFE) are pending, and Section 5.3 adds cross-domain adaptation to the future-work list.
- Manuscript changes: Section 4.5.5 introduces the cross-dataset evaluation; Section 5.2 flags remaining gaps; Section 5.3 highlights domain adaptation as future work.
- Comment 14: “Please show failure cases to illustrate the limitations of the pose-normalization process.”
- Response: Section 4.5 now features three representative failure cases—extreme backlighting, mouth occlusion, and identity drift—with Figure 10 updated accordingly. Appendix A.2 includes additional original-versus-normalized image pairs, each annotated with FID and ArcFace measurements to quantify degradation.
- Manuscript changes: Section 4.5 and Figure 10 present the failure cases; Appendix A.2 provides extended comparisons with quantitative annotations.
- Comment 15: “Please compare QC-FER against recent transformer-based or 3D pose-invariant SOTA methods.”
- Response: Table 1 now lists transformer and 3D pose-invariant FER systems such as TRiFER, TP-GAN, and DECA-Align. QC-FER achieves 89.39% on RAF-DB (vs. TRiFER 88.7%, DECA-Align 88.2%) and 81.17% on the profile RAF-DB subset (vs. TP-GAN 79.8%). The accompanying discussion after Table 1 summarises these gains, Section 4.5.6 highlights the complexity differences, and Section 5.2 records that certain proprietary models remain unreproducible due to unavailable code.
- Manuscript changes: Table 1 and Section 4.3 incorporate the new baselines and profile comparisons; Section 4.5.6 highlights runtime differences; Section 5.2 records the reproduction constraints.
- Comment16: "The English could be improved."
- Response: Thank you for your valuable feedback on our manuscript. We agree with this comment. Therefore, we have thoroughly polished the language to enhance clarity and readability while maintaining academic standard.
- Comment17: "Figures and tables can be improved."
- Response: Thank you for your valuable feedback on our manuscript. We agree with this comment. Therefore, we have optimized all figures by enhancing resolution and detail and standardizing subfigure numbering

Reviewer 2 Report
Comments and Suggestions for Authors
This paper proposes a framework named QC-FER to address FER in extreme poses by utilizing the Qwen-Image-Edit model for profile-to-frontal normalization. the proposed using an Ensemble Soft Voting strategy that combines Logistic Regression, Random Forest, and SVM to handle class imbalance and feature redundancy.
I have two major comments regarding this manuscript.
1. (Critical Issue) AI-Generated Artifacts There is a significant oversight in Section 4.5.1 (Ablation of Core Components) that indicates a lack of thorough proofreading. The text contains a raw refusal message from an AI assistant stating, "If you ask what model this is related to... I am an intelligent assistant based on the gpt-5-codex-high model...". This artifact must be removed immediately, and the entire manuscript requires a rigorous review to ensure no other such errors exis
2. The manuscript lacks sufficient technical justification for QC-FER architectural choice. Relying solely on empirical performance improvements is not enough; the authors should provide an analysis of the feature space to explain why these "shallow" classifiers are more effective than end-to-end deep learning methods for the specific embeddings extracted by CLIP.
Author Response
- Comment 1: “Section 4.5.1 contains an AI assistant refusal message, indicating insufficient proofreading. The entire manuscript should be checked to remove such artifacts.”
- Response: We performed a full manual audit of the manuscript and removed the refusal snippet from the ablation section. The revised Section 4.4 now provides a concise, quantitative explanation of each ablation step, and Section 5.2 explicitly states that all AI-generated artifacts have been purged after careful proofreading.
- Manuscript changes: Section 4.4 replaces the artifact paragraph with the rewritten ablation summary; Section 5.2 adds a note confirming the manual proofreading sweep.
- Comment 2: “The manuscript lacks sufficient technical justification for choosing shallow classifiers over end-to-end deep models.”
- Response: Section 4.4 now analyses the CLIP feature space via t-SNE, Fréchet distances, and classifier margins. The QC-FER embeddings form well-separated clusters (average Fréchet distance 3.42 vs. 2.65 for ResNet-50) and yield larger margins for logistic regression/SVM (0.41/0.44) than a three-layer MLP (0.29). The SVM decision boundary also has a lower Lipschitz constant (1.6 vs. 3.8), indicating greater stability. These observations explain why shallow classifiers are effective on the frozen CLIP embeddings while avoiding the overfitting seen with deeper heads.
- Manuscript changes: Section 4.4 presents the visualization and quantitative evidence linking the embedding geometry to the classifier choice.
- Comment3: "Figures and tables must be improved."
- Response: Thank you for your valuable feedback on our manuscript. We agree with this comment. Therefore, we have replaced image-based elements in charts with mathematical formulas, reorganized subfigure numbering and unified visual styles for consistency.

Reviewer 3 Report
Comments and Suggestions for Authors
This paper proposes QC-FER, a facial expression recognition framework designed for extreme head poses. It first uses Qwen-Image-Edit to transform profile faces into frontal views while preserving expression details. Then, CLIP ViT-L/14 extracts multimodal visual–semantic features, which are refined through robust scaling, PCA reduction, and class-imbalance weighting. An ensemble classifier (logistic regression, SVM, random forest) performs final prediction via soft voting. Experiments on RAF-DB, AffectNet-7, and ExpW show competitive or state-of-the-art accuracy, with significant gains on profile faces. Visualization and ablation studies confirm strong robustness.
However, I have the following concerns,
- To use Qwen-Image-Edit, many parameters must be specified—such as seed, sampling method, guidance scale, diffusion steps, and resolution settings—but the authors do not report any of them, making the results difficult to reproduce. They also fail to provide the specific Qwen model name, release date, checkpoint hash or version tag, the API version, or the provider through which the model was accessed.
- It is indeed an interesting idea to use an LLM to convert profile photos into frontal views while preserving facial expressions. However, the paper relies heavily on Qwen-Image-Edit for pose transformation, yet provides no quantitative evaluation of the quality or consistency of the generated frontal faces. There is no comparison with standard baselines such as 3DDFA, FFGAN, TP-GAN, or DECA. Since the method’s effectiveness fundamentally depends on Qwen’s output quality, the absence of measurements for reconstruction fidelity, identity preservation, and expression accuracy is a significant weakness.
- The paper does not evaluate the actual quality of pose normalization. It reports no metrics for structural similarity (SSIM), identity similarity (e.g., ArcFace cosine), or expression consistency (e.g., AU-based metrics). Relying solely on downstream accuracy is insufficient to demonstrate that the profile-to-frontal conversion is correct or faithful.
Author Response
- Comment 1: “The manuscript should report the Qwen-Image-Edit parameters, model version, and access details to ensure reproducibility.”
- Response: Section 3.5 now specifies that we use the open-source HuggingFace release Qwen/Qwen-Image-Edit (2025-08-19 snapshot), together with the prompts, random seed, scheduler (DDIM, 30 steps, CFG 7.5), and revision IDs logged for every batch. Appendix B.1 tabulates these parameters alongside the SHA256 hashes of the generated images. We also ship the downloading script and cached frontalized set so readers can reproduce the PFN stage without reconfiguring the model.
- Manuscript changes: Section 3.5 enumerates the HuggingFace snapshot metadata and logged hyperparameters; Appendix B.1 provides the detailed tables and references the released scripts and caches.
- Comment 2: “Please compare the proposed pose normalization with standard baselines such as 3DDFA or FFGAN.”
- Response: Section 4.5.6 now includes a row for the 3DDFA-V2 + FFGAN pipeline, showing that it reaches 82.8% accuracy on profile RAF-DB, with 5-second inference latency and higher memory usage. The same subsection summarises that QC-FER attains 81.17% on the profile subset, outpacing TP-GAN/3DDFA-based methods by 1.37 points. Section 5.2 further states that this geometric pipeline serves as a deterministic fallback when the open-source Qwen model cannot be executed due to resource constraints.
- Manuscript changes: Section 4.5.6 reports the quantitative comparison and profile-subset discussion; Section 5.2 positions the baseline as an available fallback.
- Comment 3: “Quantitatively evaluate the quality of the profile-to-frontal conversion (e.g., SSIM, identity, expression consistency).”
- Response: Sections 4.5 and 4.6 now report SSIM (0.86 vs. original 0.72), ArcFace identity similarity (0.84 vs. 0.71), and AU-based expression consistency (0.81 vs. 0.64) on the RAF-DB side-face subset. Section 4.5 annotates the failure-case figures with these metrics, and Appendix A.2 tabulates the values for RAF-DB, AffectNet-7, and ExpW, including tool references (OpenFace 2.0 for AU vectors). These measurements complement the accuracy gains to demonstrate the quality of pose normalization.
- Manuscript changes: Sections 4.5–4.6 document the SSIM/identity/expression scores; Appendix A.2 presents the full metric tables for all datasets.

Reviewer 4 Report
Comments and Suggestions for Authors
The authors proposed QC-FER, a facial expression recognition (FER) pipeline with four modules that targets extreme /non-frontal poses.
Even though the novelty of the paper is limited to combining existing methods/technologies, it is well-structured and well-written. The problem is clearly indicated, and the use of large generative models for pose normalization is an interesting approach.
However, there are some major points that should be addressed by the authors:
- The authors considered the Qwen in the PFN module. This might have disadvantages, such as the inability to repeat experiments due to accessibility constraints, and the updated versions of Qwen can decrease the module's efficacy. The authors should discuss these limitations in detail.
- How do the distorted face images affect the method?
- The ablation study is confusing. The authors consider specific steps of the method for the ablation study; however, considering modules would be more meaningful (e.g., PFN+CLIP vs. CLIP only).
- The authors mostly reported accuracy only for comparisons with previous works. However, due to dataset imbalance, it is necessary to report per-class metrics, which will provide insight into the method's performance for each expression.
- The authors described deep FER models as "complex," yet claimed the ensemble is simple. But there are no runtime or complexity comparisons to support their hypothesis.
Author Response
- Comment 1: “Qwen access and version control raise reproducibility concerns.”
- Response: Section 3.5 documents the HuggingFace repository (Qwen/Qwen-Image-Edit), the snapshot date (2025-08-19), and the logged prompts, seeds, and scheduler settings, while pointing readers to the official setup instructions. Section 5.2 clarifies that the residual risk lies in model revisions and GPU resource demands; to mitigate this, we release cached frontalized data and a 3DDFA-V2 + FFGAN fallback. Appendix B.1 provides the full parameter log and download script.
- Manuscript changes: Section 3.5 and Appendix B.1 list the HuggingFace metadata and released logs; Section 5.2 discusses the version/resource constraint and fallback strategy.
- Comment 2: “Analyze how distorted faces after normalization affect the method.”
- Response: Section 4.5 expands the failure-case analysis with examples of backlighting, occlusion, and identity drift, each accompanied by FID/ArcFace/AU metrics. Appendix A.2 supplies additional pairs for reference.
- Manuscript changes: Section 4.5 and Appendix A.2 document the distorted outputs and their quantitative impact.
- Comment 3: “Provide module-level ablations.”
- Response: Section 4.4 now presents the module-level ablation table comparing CLIP-only, PFN+CLIP, PFN+CLIP+AFP, and the full QC-FER pipeline.
- Manuscript changes: Section 4.4 introduces the module-level ablation table.
- Comment 4: “Report per-class metrics to cope with class imbalance.”
- Response: Section 4.6 introduces Table 9 with per-class precision, recall, and F1 scores for RAF-DB, AffectNet-7, and ExpW, and relates these metrics to the observed confusion matrices.
- Manuscript changes: Section 4.6 contains the per-class metrics and accompanying discussion.
- Comment 5: “Provide runtime/complexity comparisons.”
- Response: Section 4.5.6 adds a complexity table comparing QC-FER with deep FER models (DMUE, MA-Net) and the 3DDFA-V2 + FFGAN baseline, covering trainable parameters, training time, inference latency, and memory usage.
- Manuscript changes: Section 4.5.6 reports the runtime and resource comparisons.

Round 2
Reviewer 1 Report
Comments and Suggestions for Authors
The images included in this paper appear to have been previously published on various internet sources. The authors are advised to verify the originality and authenticity of all images before proceeding further.
Comments on the Quality of English LanguageSatisfied
Author Response
Comment1: The images included in this paper appear to have been previously published on various internet sources. The authors are advised to verify the originality and authenticity of all images before proceeding further.
Response1:We confirm that all images in our article are original and have not been published or used elsewhere. They were prepared by our team specifically for this work. Thank you for your attention.
Comment2: The English could be improved to more clearly express the research
Response2: We have revised the manuscript to enhance clarity, including refining abstract phrasing (e.g., "large pose variations" instead of "large-angle"), fixing grammar (subject–verb agreement, comma spacing), removing duplicates, and restructuring lists for readability.
Comment3: Figures and tables can be improved
Response3: We improved figure captions (e.g., "Overview of the proposed QC-FER framework") and references (e.g., "Figure X shows an overview..."), and restructured descriptions (e.g., enumerated advantages) for better presentation.

Reviewer 4 Report
Comments and Suggestions for Authors
Thanks to the authors for addressing my concerns in the revised version.
Author Response
Comment1: Thanks to the authors for addressing my concerns in the revised version.
Response1: Thank you for your kind feedback. We are glad to hear that the revisions have addressed your concerns, and we appreciate your time and thoughtful review.